# Nontrivial band geometry in an optically active system

Jiahuan Ren [1,2], Qing Liao [1✉], Feng Li [3,4✉], Yiming Li[3], Olivier Bleu[5], Guillaume Malpuech[5], Jiannian Yao[2], Hongbing Fu [1,2✉] & Dmitry Solnyshkov [5,6✉]

Optical activity, also called circular birefringence, is known for two hundred years, but its applications for topological photonics remain unexplored. Unlike the Faraday effect, the optical activity provokes rotation of the linear polarization of light without magnetic effects, thus preserving the time-reversal symmetry. In this work, we report a direct measurement of the Berry curvature and quantum metric of the photonic modes of a planar cavity, containing a birefringent organic microcrystal (perylene) and exhibiting emergent optical activity. This experiment, performed at room temperature and at visible wavelength, establishes the potential of organic materials for implementing non-magnetic and low-cost topological photonic devices.

[1] Beijing Key Laboratory for Optical Materials and Photonic Devices, Department of Chemistry, Capital Normal University, 100048 Beijing, China. [2] Tianjin Key Laboratory of Molecular Optoelectronic Sciences, Department of Chemistry, School of Sciences, Tianjin University, Collaborative Innovation Center of Chemical Science and Engineering, 300072 Tianjin, China. [3] Key Laboratory for Physical Electronics and Devices of the Ministry of Education & Shaanxi Key Lab of Information Photonic Technique, School of Electronic and Information Engineering, Xi'an Jiaotong University, 710049 Xi'an, China. [4] Department of Physics and Astronomy, University of Sheffield, Sheffield, UK. [5] Institut Pascal, PHOTON-N2, Université Clermont Auvergne, CNRS, SIGMA Clermont, F-63000 Clermont-Ferrand, France. [6] Institut Universitaire de France (IUF), 75231 Paris, France. ✉email: liaoqing@cnu.edu.cn; felix831204@xjtu.edu.cn; hbfu@cnu.edu.cn; dmitry.solnyshkov@uca.fr

The exploration of photonic systems involving the concepts of topology and non-reciprocity has become a mainline of scientific activity in recent years, driven by both funda-mental and applied motivation. Indeed, the implementation of microscopic optical isolators is absolutely crucial for the devel-opment of integrated photonics[1] and robust quantum optical circuits[2]. Photonic topological insulator analogs[3–5] and topolo-gical lasers[6–10] with topological edge modes represent a solution to this stringent request. There are two great families of 2D topological insulators[11]. One is based on the Quantum Hall effect (QHE), either normal or anomalous (QAHE), in systems with broken time-reversal symmetry (TRS). The energy bands possess non-zero integrated Berry curvature (Chern numbers), which leads to non-reciprocal transport on the edges. Of course, QAHE is not limited to photonics: it has been originally proposed[12] and recently demonstrated in electronics[13,14] and atomic lattices[15]. The other family is based on the Quantum Spin Hall effect (QSHE). The total band Chern number is zero, but the band geometry remains non-trivial. In particular, it is possible to separate two spin or pseudo-spin domains, each being char-acterized by a non-zero Berry curvature, opposite between the two. The integration over the (pseudo)-spin subbands gives non-zero (pseudo)-spin Chern numbers. If the corresponding (pseudo)-spins are protected by a symmetry, such as the TRS, which protects the electron's spin, non-reciprocal (pseudo)-spin transport on the edge or interface states can take place. This type of effect has been demonstrated in photonics with various types of pseudo-spin realizations, each being approximately protected by a specific symmetry[16–20].

Photonic QAHE requires the combination of photonic spin-orbit coupling (SOC)[21], which is an intrinsic property of 2D confined photonic media like waveguides and planar cavities, with TRS breaking by the Faraday effect[22] induced by an applied magnetic field. Because of these two contributions, the photonic modes of a 2D continuous medium exhibit a non-zero Berry curvature[23,24], with the possibility to define the associated topo-logical invariants, such as the Chern numbers[25] and the $\mathbb{Z}_2$ invariant[26], and to observe the edge states and a non-zero angular momentum[27]. Once inserted in an appropriate 2D lattice, these modes demonstrate topological gaps and non-reciprocal trans-port on the lattice edge[3,4,7,9,28,29]. However, the Faraday effect is usually small at optical wavelengths. It requires large magnetic fields, hindering practical applications. The so-called optical activity (OA) is another type of optical response discovered at the beginning of the XIXth century[30,31], leading (similar to the Faraday effect) to the rotation of the linear polarization of light during its propagation. Unlike the Faraday effect, OA is an intrinsic property linked with the chirality of a structure. It does not require magnetic field and preserves the TRS, as sketched in Fig. 1. In the Faraday effect (Fig. 1a), the angle of rotation of the polarization $\alpha_0$ continues to increase for inverted propagation direction. With OA (Fig. 1b), the polarization starts to rotate backwards to its original position, thus preserving the TRS. Because of this symmetry property, OA does not allow to obtain non-zero Chern numbers and thus cannot be used to obtain the QAHE. However, it does lead to a local non-zero Berry curvature of the bands, and thus it can lead to the anomalous Hall effect, important for optovalleytronics.

The OA can arise from the presence of chirality at different scales: from the structure of single molecules or crystal unit cells[32,33], from the stacking of monolayers[34], or, finally, at the macroscopic scale of the structure as a whole. The latter config-uration has been demonstrated recently in a cavity exhibiting a degeneracy of orthogonally polarized modes of opposite parity[35], as shown in Fig. 1c, d. In this regime, the active region behaves as a $\lambda/2$ plate, which is known to invert the Stokes polarization

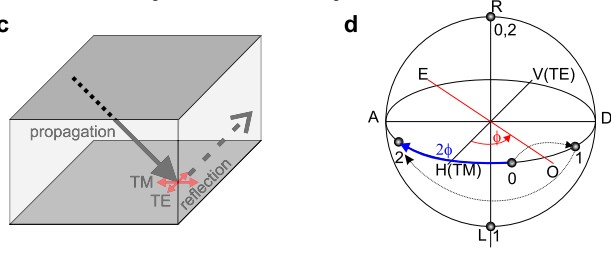

Faraday effect and optical activity: Classical configuration

Emergence of OA from birefringence in cavities

**Fig. 1 Faraday effect and optical activity.** Scheme showing the rotation $\alpha_0$ of the linear polarization (red double arrow) of light while making a back and forth trip, in the Faraday effect (**a**, TRS broken by magnetic field $\vec{B}$ - green) and in a chiral OA structure (**b**, TRS not broken, screw axis in blue). Black arrows indicate the propagation direction. **c** Propagation and reflection of light in a cavity with a birefringent active region; **d** Poincaré sphere showing the polarization states ("0" - initial, "1" - after propagation, "2" - after reflection) and the inversion axes (TE-TM - transverse-electric/transverse-magnetic, OE - ordinary-extraordinary, rotated by $\phi$), as well as the polarization basis (H-horizontal, V - vertical, D - diagonal, A -anti-diagonal, R - right-circular, L - left-circular). After propagation and reflection, linear polarization is rotated by $2\phi$.

vector of a propagating beam with respect to the ordinary-extraordinary axis (marked "OE" on the Poincaré sphere in Fig. 1d). The initial polarization of the beam is marked "0", and the state after the propagation is marked "1". But the reflection on a metallic mirror also leads to the polarization inversion with respect to a different axis (TE-TM or HV in this case, also marked in Fig. 1c, d), except the normal incidence case, which gives the final polarization state marked "2". Mathematically, two inver-sions (black dashed arrows) are equivalent to a rotation (blue arrow) by the double of the angle $\phi$ between the axes of inversion (also marked in Fig. 1d), which means that any linear polarization is not an eigenstate of this system for a beam with non-zero in-plane wave vector. On the contrary, the circular polarization, inverted twice, recovers the original state ("2"="0", marked in Fig. 1d), which shows that the eigenstates are circular-polarized. The details of the derivation can be found in Rechcińska et al.[35] and in Methods section. Such degeneracy requires a strong birefringence of the active region of the cavity in order to split the polarization eigenmodes of the same parity and bring them close to the modes of opposite parity. This birefringence can be pro-vided by organic materials, for example, perylenes ($C_{20}H_{12}$)[36]. Perylenes are nowadays widely used in optics due to their intense light absorption and luminescence in the visible range, high sta-bility, and quantum yield[37]. Their optical properties are pro-mising for solar cells[38], energy harvesting, temperature control[39], lasers[40], and other nanophotonic applications[41]. Perylenes are also used in the field of 2D materials to hold together van der Waals heterostructures[42] or by themselves[43].

In this work, we establish the potential of OA structures for topological photonics. We study a basic photonic element – a planar cavity with a birefringent crystal, exhibiting chirality at the

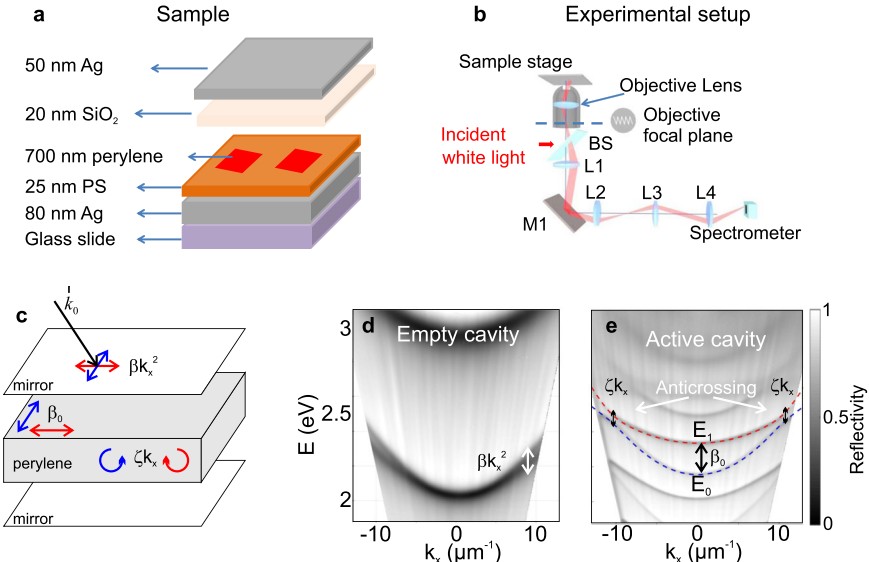

**Fig. 2 Combining the spin-orbit couplings. a** The sample consists of a perylene crystal embedded in a microcavity. **b** Experimental setup allowing to obtain polarization-resolved complete state tomography. BS: beam splitter; L1-L4: lenses; M1: mirror. The red beam traces the optical path of the reflected light from the sample at a given angle. **c** Embedding the perylene crystal in a planar cavity combines TE-TM splitting $\beta k^2$, linear birefringence $\beta_O$, and emergent OA $\zeta k_x$. **d** Reflection of an empty cavity versus energy and wave vector, which evidences the k-dependent TE-TM splitting $\beta k^2$. **e** Reflection of a cavity filled with perylene, which is inducing linear birefringence $\beta_O$ on top of the cavity-induced TE-TM splitting, leading to emergent OA $\zeta k_x$ (black arrows). Dashed lines show a fit of the two coupled bands with the Hamiltonian (4). The axes of the polarizer and the excitation direction $k_x$ are aligned with the fast axis of the crystal. The figures are measured for a fixed angle range; white regions appear due to angle-to-wave vector conversion.

macroscopic scale. We provide a direct measurement of the complete band geometry, namely the Berry curvature and the quantum metric of the photonic bands. We show that their non-trivial distribution can be interpreted with a two-by-two effective Hamiltonian accounting for the interplay of the polarization effects involved.

## Results

**Photonic spin–orbit coupling**. We start by embedding the perylene crystal within a metallic microcavity (Fabry–Perot resonator) as sketched in Fig. 2a. The quantization of the z-component of the wave vector leads to the formation of 2D bands parametrized by the in-plane 2D wave vector $\mathbf{k} = (k_x, k_y)^T$. The modes we consider are above the light cone. They have small in-plane wavevector, parabolic dispersion, and are radiatively coupled to the outside of the cavity[44]. The experimental setup (see Methods for details), allowing to make a full optical state tomography out of which the band geometry (Berry curvature and quantum metric) can be reconstructed[23], is shown in Fig. 2b. All experiments are performed at room temperature. The reflection of the sample is measured versus energy and in-plane wave vector for the six different light polarizations (left and right circular, horizontal-vertical, and diagonal-anti-diagonal), which allows a full determination of the three Stokes vector components (light polarization pseudo-spin) of the modes. The axes of the polarizer are aligned with the crystal axes and with the axes of the reciprocal space defined by the CCD camera. The orientation of the Stokes vector $\mathbf{S}(\mathbf{k})$ on the Poincaré sphere is given by the polar angle $\theta(\mathbf{k})$ and the azimuthal angle $\phi(\mathbf{k})$. The measurement of these quantities allows to extract the Berry curvature[45] $B_z$ and the quantum metric[46] $g_{ij}$ of a given mode as[47]:

$$g_{ij} = \frac{1}{4}(\partial_{k_i}\theta\partial_{k_j}\theta + \sin^2\theta\partial_{k_i}\phi\partial_{k_j}\phi) \quad (1)$$

$$B_z = \frac{1}{2}\sin\theta(\partial_{k_x}\theta\partial_{k_y}\phi - \partial_{k_y}\theta\partial_{k_x}\phi) \quad (2)$$

The Stokes vector can be found as an eigenstate of an effective $2 \times 2$ Hamiltonian, accounting for the polarization effects arising in the structure. It can describe two modes which are close to each other, whether they are of the same parity (as in inorganic microcavities) or not (as is the case here). To facilitate the understanding, we decompose the combination of polarization effects into a set of individual contributions. The first contribution is the TE-TM splitting, ubiquitous in 2D photonic systems[24]. In general, the TE-TM splitting appears in any inhomogeneous system, in presence of any gradient allowing to define the transverse directions for the field[32]. In particular, in planar cavities it appears because of the polarization-dependent reflection coefficients[48]. In an ideal case, the above-mentioned quantized modes of an empty cavity are doubly polarization degenerate at zero in-plane wavevector. This polarization degeneracy is lifted by different contributions sketched in Fig. 2c. The TE-TM splitting of the bare cavity[21,48] characterized by $\beta$ is zero at $k = 0$ and then grows quadratically with $k$:

$$E_{TE,TM} = \frac{\hbar^2 k^2}{2m_{TE,TM}} = \frac{\hbar^2 k^2}{2m} \pm \beta k^2 \quad (3)$$

with $m_{TM}$ and $m_{TE}$ corresponding to the longitudinal and transverse effective masses $(m^{-1} = (m_{TE}^{-1} + m_{TM}^{-1})/2$ and $\beta = \hbar^2(m_{TE}^{-1} - m_{TM}^{-1})/4)$. The parameter $\beta$ thus describes the cavity TE-TM splitting. The difference of this splitting for the modes of different order that we consider can be neglected, because this order is sufficiently high and thus the variation of all parameters with the mode number is small. The second contribution is the linear birefringence of the perylene crystal, described by $\beta_0$, which splits the linearly polarized different-parity modes H and V at $k = 0$: $E_{H,V}(k = 0) = \pm\beta_0$. We note that $\beta_0 = 0$ does not mean zero birefringence, on the contrary, it means that the polarization splitting for two modes of the same order is equal to the splitting between the modes of the same polarization and

 

different order (sometimes called free spectral range). The important difference between the TE-TM and the H–V splittings is that the orientation of TE-TM is defined by the in-plane wave vector (the orientation of the incidence plane with respect to the polarization-resolving detector), while the orientation of the H–V splitting is linked with the crystal axes and thus stays constant, because the sample is not rotated in our experiments.

The two contributions introduced above cancel each other at two points $\pm k_0 = \sqrt{\beta_0/\beta}$ along the $k_x$ axis. Together, they determine the linear polarization of the modes ($S_1$, $S_2$ Stokes components). Finally, the third contribution is the emergent optical activity (chirality) of the structure, described by the parameter $\zeta$. As shown in Rechcińska et al.[35], for almost degenerate modes of opposite parity, the induced chirality appears as a splitting between the circular-polarized modes, linear in $k_x$:

$$E_\pm = E_0 \pm \zeta k_x \qquad (4)$$

It therefore determines the circular polarization degree of the modes ($S_3$). We note that the form of this term does not depend on the origin of the optical activity: because of the time-reversal symmetry, the expression must be an odd power of a certain in-plane wave vector projection. All these effects are combined in an effective $2 \times 2$ Hamiltonian describing the polarization eigenstates. We write it on the circular polarization basis:

$$H_{\mathbf{k}} = \begin{pmatrix} \frac{\hbar^2 k^2}{2m} + \zeta k_x & \beta_0 + \beta k^2 e^{2i\varphi} \\ \beta_0 + \beta k^2 e^{-2i\varphi} & \frac{\hbar^2 k^2}{2m} - \zeta k_x \end{pmatrix} \qquad (5)$$

where $\varphi$ is the polar angle. The energy dispersion of this Hamiltonian as a function of wave vector $E(\mathbf{k})$ is shown in Fig. 2e with dashed lines. As any $2 \times 2$ Hermitian Hamiltonian, it is a linear combination of Pauli matrices that can be physically interpreted as an effective magnetic field acting on the Stokes vector, that is, $H_{\mathbf{k}} = H_0 + \mathbf{\Omega} \cdot \boldsymbol{\sigma}$, where $H_0$ is the diagonal kinetic energy part, $\boldsymbol{\sigma}$ is a vector of Pauli matrices and $\mathbf{\Omega}$ is the effective field, which reads:

$$\mathbf{\Omega}(\mathbf{k}) = \begin{pmatrix} \beta_0 + \beta k^2 \cos 2\varphi \\ -\beta k^2 \sin 2\varphi \\ \zeta k_x \end{pmatrix} \qquad (6)$$

As defined above, $\beta_0$, $\beta$, and $\zeta$ determine the strength of the effective fields corresponding respectively to linear birefringence, the k-dependent TE-TM splitting, and emergent OA, which can be viewed as an effective Zeeman splitting. The k-dependent effective fields can both be interpreted as photonic SOCs. The Stokes vector of the eigenmodes is either aligned or anti-aligned with the effective field $\mathbf{\Omega}$. This picture allows to find these modes easily and to predict their Berry curvature and quantum metric. In vicinity of the points $k_x = \pm k_0$, the Hamiltonian (5) can be written as a Rashba Hamiltonian with a constant Zeeman splitting, which was shown to be equivalent to a Hamiltonian of a non-relativistic quantum particle coupled to a non-Abelian Yang-Mills field[49–51].

Figure 2d shows the total reflection coefficient of an empty cavity with bare TE and TM modes with different effective masses (because of $\beta$), but degenerate at $k = 0$. The cavity filled with the active material (Fig. 2e) shows a radically different mode dispersion. The first visible consequence is that the cavity is optically thicker due to the perylene crystal refractive index ($n \approx 2$). More importantly, the splitting between linearly polarized modes at $k = 0$ becomes comparable with that of the modes of different orders, because of the significant linear birefringence of perylene (refractive indices 1.7 and 2.5[52]). These modes anticross at a finite wave vector (instead of simply crossing)

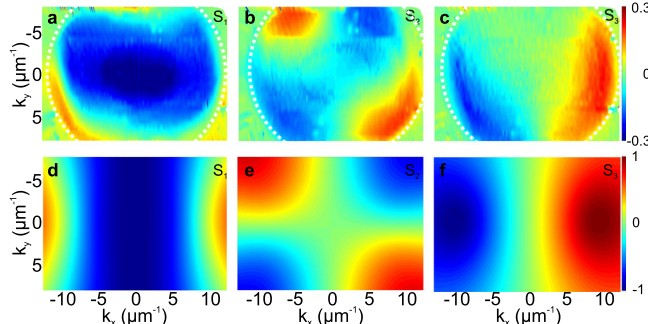

**Fig. 3 Experimentally measured and theoretically calculated Stokes parameters.** Measured Stokes parameters of the mode $E_0$ from Fig. 2e: **a** $S_1$, **b** $S_2$, **c** $S_3$. The boundary of the meaningful (non-zero) signal is shown with a white line; Theoretically calculated Stokes parameters for the lowest eigenstate of (5): **d** $S_1$, **e** $S_2$, **f** $S_3$.

because of the emergent OA $\zeta$ which can be associated with an effective optical activity coefficient: $\alpha \approx \zeta k_x n^2/2hc$ (see Methods for more details and Supplementary Fig. 6 for the polarization-resolved image of the dispersion). The key difference with respect to a cavity with TRS broken by the Faraday effect[24] is that here, the effective Zeeman field changes sign with $k_x$. This Hamiltonian shows two gapped tilted Dirac cones at the two reciprocal space points where the in-plane components of the field cancel. The sign of the mass term, opposite for the two cones, is given by the sign of the effective Zeeman field. We note that Fig. 2e corresponds to the direction with the smallest gap, $k_x$. We have mapped the whole reciprocal space and we can therefore exclude the possibility that the observed anticrossing is simply due to a tilt of an optical axis which could shift the crossing point away.

**The Stokes vector and the quantum geometric tensor.** The validity of the effective Hamiltonian is confirmed by the measured 2D wave vector maps of the Stokes vector components of the lower branch, shown in Fig. 3a–c compared with theoretical predictions shown in panels (d–f). We note that the experimentally measured Stokes components are zero outside an elliptic region where the detection is efficient. Inside this region (marked with a white dashed line), the experiment and the theory exhibit a good agreement. As expected, the linear birefringence is compensated by the k-dependent TE-TM field at the anticrossing points[53]. The two components $S_1$ and $S_2$ of the Stokes vector cancel and change sign around these points, forming 2D monopoles. The third Stokes vector $S_3$ component is maximal at these two points but is changing sign at $k_x = 0$ because of the TRS. The cross-sections of the dispersion close to the anticrossing points together with the pseudospin orientation for the two branches are shown in Supplementary Fig. 1. The opposite circular polarization of the branches is also directly visible in Supplementary Fig. 6, showing the circular polarization degree of the reflection at different energies without any extraction.

The measurements of the Stokes vector allow to extract the Berry curvature and the quantum metric of the modes, as shown in Fig. 4a, b. As expected from the TRS (encoded in the Zeeman SOC and the $S_3$ texture), the Berry curvature shows two maxima of opposite signs, which means that the integrated Berry curvature over the band is zero. However, by separating the reciprocal space in two regions, like in the quantum valley Hall effect[20], it is possible to associate non-zero pseudo-spin Chern numbers to each of these two regions. These regions can viewed as being analogs of valleys, which emerge in this optically active system without the need of using a lattice. The trace of the quantum metric (Fig. 4b) is maximal in the regions of the anticrossing, where the Stokes vector rotation is

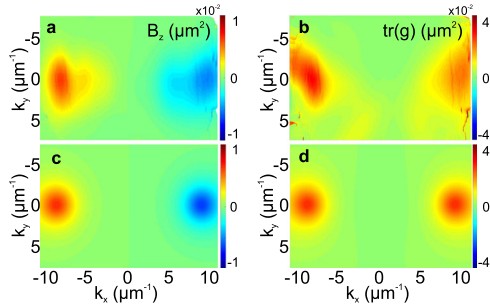

**Fig. 4 Experimentally extracted and theoretically calculated Berry curvature and quantum metric.** Quantum geometry extracted from the measured Stokes vector: **a** Berry curvature $B_z$, **b** trace of the quantum metric $g_{xx} + g_{yy}$; Calculated quantum geometry for the lowest eigenstate of (5): **c** Berry curvature $B_z$, **d** trace of the quantum metric $g_{xx} + g_{yy}$.

the fastest. The experimentally extracted geometry (a,b) corresponds very well to the theoretical predictions (c, d) based on the $2 \times 2$ effective Hamiltonian (5) with parameters $\beta_0 = 0.18$ eV, $\beta = 9 \times 10^{-4}$ eV$\mu$m$^2$, $\zeta = 2.5 \times 10^{-3}$ eV$\mu$m, extracted from the experimental dispersion (Fig. 2e). The distortion of the maxima of the Berry curvature in experiment could be explained by the contribution of the exciton resonance (see Methods section and Supplementary Fig. 3) and by the difference of the experimental resolution in the two directions. The OA coefficient obtained from the splitting at the anticrossing point demonstrates a remarkably high value of $\alpha = 1.4 \times 10^4$ degrees/mm. This is a crucial ingredient which has allowed room temperature measurements. Indeed, for a typical OA material such as the tartaric acid, the mode splitting at the anticrossing point would be of the order of $10^{-4}$ meV, much smaller than the broadening $k_B T_{RT} \approx 26$ meV. The room temperature operation at optical wavelengths is highly favorable in the prospect of using such non-trivial band geometry for implementing practical topological photonic devices.

## Discussion

While our study belongs to the field of classical optics, since we are dealing with classical photonic beams, it can nevertheless have important implications for quantum mechanics. Indeed, it is well known that the transverse behavior of a light beam in the paraxial approximation is well described by the Schrödinger equation, with the propagation axis playing the role of time. The same applies to planar cavities, but with the meaning of the temporal axis restored. Our system represents therefore a model of a quantum system in many senses. The Stokes vector of light is an equivalent of the spin of an electron, and the non-zero Berry curvature of both the Poincaré sphere and the Bloch sphere is the most direct consequence of this analogy. The degrees of freedom provided by the direct and the reciprocal space are also equivalent to quantum mechanics, which allows to use both the languages of Maxwell and Schrodinger equations for topological photonics.

The original antisymmetric distribution of the Berry curvature that we have observed for an OA system crucially affects the numerous phenomena driven by the band geometry, the emblematic one being the anomalous Hall effect family which includes valley Hall effects at the heart of valleytronics[54]. With broken TRS (same-sign Berry curvature), the anomalous Hall drift does not change sign upon time reversal (like the Faraday rotation angle $\alpha_0$). With conserved TRS (opposite Berry curvature for $\pm k_x$), the anomalous Hall drift is reversed and the system returns to its original position upon time reversal. These valleys can be selectively excited by simply controlling the beam incidence angle. The precision of such control is very high, since each valley spans $\sim 10°$, while a typical beam spans $\sim 4$ arc minutes.

Another important quantity which our measure allows to access is the quantum metric[46]. Quantum metric has been recently found to be associated with many phenomena and it became a hot research topic[55]. In optics, it allows to quantify the non-adiabaticity of realistic transport experiments[23], which is certainly crucial to operate devices based on geometrically non-trivial bands such as valleytronic or opto-valleytronic systems. The large value of the emergent OA coefficient provides a strong protection against the non-adiabaticity, allowing to use very high spatial gradients: the maximal anomalous Hall drift of 0.6 μm for our parameters can be achieved at a propagation distance of only 45 μm.

Quantum geometry is currently a subject of active studies. It is studied both globally, at the level of topological invariants, and locally, as a distribution of the Berry curvature and quantum metric in a certain parameter space. Topological invariants with their discrete integer values and associated global effects, such as the presence of edge states determined via the bulk-edge correspondence, are easier to be measured experimentally[56–59]. The local distribution of the Berry curvature and, later, the quantum metric, have usually been measured via the related dynamical effects[60–66], such as the anomalous Hall drift. Recently, the quantum geometry has been extracted from the eigenstates of a photonic system, like in the present work, with an additional confirmation of the results by the anomalous Hall measurements[24].

With the Berry curvature localized in analogs of valleys which are well represented mathematically by tilted Dirac cones, it could be possible to create interface states of the Jackiw-Rebbi type[67] between regions of opposite topology defined by the sign of the emergent optical activity $\zeta$, with a single chiral state for each "valley". These states exhibit valley-dependent group velocity, whose direction (in the reference frame of the tilted Dirac cone) can be inverted by changing the order of the topological materials, as in the quantum valley Hall effect[17,68]. This behavior, confirmed by our preliminary simulations, will be a subject of a separate future work.

One more interesting outcome of our work is the possible implementation of an artificial magnetic field acting on photons[35]. Indeed, the effect of the OA in the Hamiltonian (4) with zero $\beta_0$ can be represented as an action of a vector potential, opposite for the two spin components ($\pm$)

$$\hat{H}^{\pm} = \frac{1}{2m}(\hat{\mathbf{p}} - e\mathbf{A}^{\pm})^2 \qquad (7)$$

where the vector potential is given by $A_x^{\pm} = \pm m\zeta/e$. Making the in-plane OA position-dependent, for example $\zeta(y) = \zeta_0 y$, leads to the emergence of a magnetic field $\mathbf{B}^{\pm} = \nabla \times \mathbf{A}^{\pm}$ of opposite sign for the two circular polarizations. This field gives rise to the formation of Landau levels or Harper-Hofstadter-like energy spectra in periodic structures. We note that this field is Abelian, in spite of being spin-dependent. This synthetic magnetic field is similar in spirit to the one realized in refs. [16,35], but with the use of the intrinsic polarization pseudospin, which does not require the presence of an artificial lattice. In order to induce such spatial dependence, one can introduce a slight variation of the background refractive index in the cavity or of the cavity thickness, which then affects the $\zeta$ contribution in the Hamiltonian (1). Tuning the coefficients of the Hamiltonian by choosing different materials would allow to deeply modify the band geometry. In Gianfrate et al.[24], a system described by a similar effective Hamiltonian was studied, except that the OA was replaced by an effective Zeeman splitting. The bands were showing two split Dirac cones, like in the present work, but with the same sign of the Berry curvature in a given band. By tuning the linear birefringence $\beta_0$ to zero in Eq. (5), the dispersion bands would exhibit

a crossing at $k = 0$, but the reciprocal space nevertheless remains split into two "valleys" of opposite Berry curvature because of the OA. Instead of being concentrated at the anticrossing points, the Berry curvature exhibits a crescent shape in this case. If both OA and birefringence are set to zero, which is the case of an empty cavity shown in Fig. 2d, the dispersion represents two touching parabola, which are characterized by two Berry monopoles of opposite charges at $k = 0$.

Our results show that the organic microcrystals, such as perylene, which are promising for the implementation of classical photonic elements because of their remarkable basic optical properties, are also appealing to implement topological photonic devices, because such devices do not necessarily require broken TRS[69]. The polarization tomography measurements allowed us to fully characterize the quantum geometry of photonic bands. These measurements have revealed that the optical activity of a system, independent of its origin, renders the TRS photonic bands geometrically non-trivial, exhibiting gapped Dirac cones with non-zero Berry curvature. The precise measurements of the quantum geometry of these bands favor the development of quantitative optovalleytronics.

## Methods

**Material structure and fabrication**. The perylene (99%+) and TBAB (Tetrabutylammonium bromide) used in the experiment were purchased from Acros and Innochem respectively without further purification. The 2D square sheet of crystalline perylene was prepared by space-confined strategy[70]: 50 μL of 0.5 mg/ml perylene/chlorobenzene was first added onto 1mg/ml TBAB/water solution. After the complete evaporation of chlorobenzene, 2D square sheets of crystalline perylene were formed on the solution surface, exhibiting a thickness of 200–1000 nm. The molecule arrangement of the perylene film is illustrated in Supplementary Fig. 2.

**Cavity structure**. For the empty microcavities characterized by Fig. 2d, 80 nm silver film was first evaporated on a glass substrate, followed by a spin-coating of 300 nm polystyrene (PS) film and a final vacuum evaporation of 50 nm silver film. The reflectance of the 80-nm and 50-nm silver films were 99.4% and 87% respectively, enabling easier light extraction from the top mirror of the cavity. For the active microcavity embedding perylene, 25 nm PS film was first spin-coated on 80 nm silver film, then the prepared 2D perylene sheets (thickness ~ 750 nm for the studied cavity) were transferred onto the PS film by bringing them in contact at the surface of the TBAB/water solution. A final evaporation of 20 nm SiO₂ and 50 nm silver film was made to form the microcavity sketched in Fig. 2a.

**Spectroscopy**. The angle-resolved spectroscopy was performed at room temperature by the Fourier imaging using a ×100 objective lens of a NA 0.95, corresponding to a range of collection angle of ±70°. As sketched in Fig. 2b, an incident white light from a Halogen lamp was focused on the area of the microcavity containing perylene, and the k-space or angular distribution of the reflected light was located at the back focal plane of the objective lens. Lenses L1-L4 formed a confocal imaging system together with the objective lens, by which the k-space light distribution was first imaged at the right focal plane of L2 through the lens group of L1 and L2, and then further imaged, through the lens group of L3 and L4, at the right focal plane of L4 on the entrance slit of a spectrometer equipped with a liquid-nitrogen-cooled CCD. The use of four lenses here provided flexibility for adjusting the magnification of the final image and efficient light collection. Tomography by scanning the image (laterally shifting L4) across the slit enabled obtaining spectrally resolved 2D k-space images. In order to investigate the polarization properties, we placed a linear polarizer, a half-wave plate and a quarter-wave plate in front of the spectrometer to obtain the polarization state of each pixel of the k-space images in the horizontal-vertical (0° and 90°), diagonal (±45°), and circular ($\sigma^+$ and $\sigma^-$) basis[71,72].

**Extraction of the Stokes parameters and the quantum geometry**. The experimental tomography images represent a set of intensities in six polarization components measured in reflection as a function of in-plane wave vector ($k_x$, $k_y$) and wavelength $\lambda$. To obtain the maps of the Stokes vector components for a given mode, shown in Fig. 3a–c, we proceed as follows. We consider a reflectivity spectrum measured under white light excitation for total intensity, such as shown in Supplementary Fig. 7 (black circles) for a single point of the reciprocal space. We first determine the wavelength $\lambda_0$ and the energy $E_0$ corresponding to the particular mode, by fitting the total reflection spectrum with Lorentzian-broadened resonances over an approximately linear background (red solid line). We then fit the individual intensity components to determine the relative weight of resonance

(taking into account the magnitude and the width of the peak) in each of the six polarizations (blue and violet triangles in Supplementary Fig. 7 for experimental circular polarization and red dashed and dash-dotted lines for theory), which allows finally to determine the three components of the Stokes vector. In our example, we show only two polarization projections of the six (to avoid overloading the figure). We note that the positions of the reflectivity minima detected in two polarizations under a non-polarized excitation do not necessarily correspond to the positions of the modes: their position depends on the linewidth and the polarization degree, and the maximal deviation can be of the order of the linewidth. As an example, the reflectance in the two circular polarizations is given by:

$$R_\pm(E) = 1 - \frac{\left(E - \frac{\hbar^2 k^2}{2m} - (\beta_0 - \beta k^2) \mp \zeta k\right)^2 + \Gamma^2}{\left(E^2 - \left(\frac{\hbar^2 k^2}{2m}\right)^2 - (\beta_0 - \beta k^2)^2 - (\zeta k)^2 - \Gamma^2\right)^2 + (2E\Gamma)^2} \quad (8)$$

where $\Gamma$ is the linewidth, and all other terms have been introduced in the Hamiltonian (5).

The quantum geometric tensor components $g_{ij}$ and $B_z$ are extracted from the Stokes vector according to Eq. (1) of the main text. Lowpass Fourier-transform smoothing is applied to the maps of the angles $\theta$ and $\phi$ before calculating the partial derivatives numerically.

**Coupling with excitons**. The high exciton oscillator strength in perylene is known[52] to cause the strong coupling effect: in vicinity of the exciton resonance, the eigenstates are not purely photonic or excitonic, but become a mixture of the two. The strong coupling is characterized by the Rabi splitting, which is the splitting of the mixed exciton-polariton modes. In ref. [52], it has been estimated as $V_R = 140$ meV, more than enough for the observation of the effect at room temperature.

There are numerous consequences of the strong coupling: the non-parabolicity of the polariton dispersion and the dependence of the polarization splittings on the wave vector. While a detailed study of the strong coupling in a cavity with perylene is beyond the scope of the present work, we note that these effects do show up in our experimental measurements. The dispersions shown for the cavity with an active region deviate from parabolicity for high wave vectors, and the observed distribution of the Berry curvature and the quantum metric deviates from the predictions of a simple model based on a $2 \times 2$ effective Hamiltonian. The strong coupling could be one of the possible reasons of these deviations, which we demonstrate in the Supplemental Fig. 3, calculated with a $4 \times 4$ Hamiltonian taking into account the strong coupling of the excitonic and photonic states. Indeed, this figure exhibits a better agreement with the experiment (Fig. 4a) thanks to the elongated shape of the modes. We note, however, that all the essential physics of the system is already captured by the $2 \times 2$ effective Hamiltonian, which is why we have used it throughout in our work.

**Optical activity**. Different formalisms have been developed for the description of the optical properties of crystals. We start with the equation of the optical indicatrix or the index ellipsoid[32,33], obtained from the relations between $\mathbf{D}$ and $\mathbf{E}$ in the medium, which for a crystal with indices $n_1$ and $n_2$ in the absence of the OA reads

$$(n^2 - n_1^2)(n^2 - n_2^2) = 0 \quad (9)$$

where $n$ is the refractive index for a given direction. For an OA medium, the optical gyration vector is introduced as a small correction to this equation:

$$(n^2 - n_1^2)(n^2 - n_2^2) = G \quad (10)$$

Near an optical axis, where $n_1 \approx n_2$, this equation can be rewritten in the first order as

$$n_\pm = \bar{n} \pm \frac{G}{2\bar{n}} \quad (11)$$

where $\bar{n}$ is the average refractive index, whereas $n_+$ and $n_-$ are the refractive indices for the two circular polarizations. The same result can be obtained using various other formalisms, for example using the Berreman matrices[73]. The difference of the refraction indices leads to the rotation of the polarization plane of linearly polarized light, which is the most well-known signature of OA. This rotation is usually characterized by the optical activity coefficient $\alpha$ expressed in degrees/mm or in rad/mm. Its link with difference in the refractive indices $n_+$ and $n_-$ is given by the formula $\Delta n = \alpha \lambda / \pi$ where $\lambda = 2\pi/k_0$. Here, $k_0$ is the total wavevector of light composed of its in-plane ($k$, used in the main text) and vertical ($k_z$) projections $k_0 = \sqrt{k_z^2 + k^2}$. The OA coefficient can be written in terms of the gyration vector $G$ as $\alpha = \pi G / \lambda \bar{n}$ and in terms of the gyration tensor $\eta_{ij}$ (usually called $g$, but here we use $\eta$ not to be confused with the quantum geometric tensor of the main text) as $\alpha = \pi \eta_{ij} N_i N_j / \lambda \bar{n}$ ($N_i$ are the direction cosines). These different representations of OA are used in different fields: the OA coefficient $\alpha$ is used to characterize the angle of rotation in the transmission configuration, whereas the gyration tensor $\eta$, being a part of the dielectric permittivity tensor $\epsilon$, is used for the calculation of confined optical modes, like in our case.

The gyration tensor value corresponding to the experimentally measured dispersion is $\eta_{xz} \approx 0.14$, much larger than any of the tensor components of quartz $\eta_Q \sim 10^{-5}$, which guarantees that our observations are not due to the optical activity of quartz, a layer of which is present inside the microcavity. At the anticrossing wavelength $\lambda \approx 500$ nm, the splitting between the energies is $E_+ - E_- \approx 50$ meV, which gives an equivalent value of $\alpha = 500$ rad/mm or $\alpha = 2.8 \times 10^4$ degrees/mm. This is comparable with the values observed in metamaterials, such as chiral photonic crystals[74]. It starts to approach the optical activity of chiral stacks of 2D materials, where the rotation of tens of degrees can be observed for only 10 monolayers of material, and the corresponding $\alpha_{chir} \sim 5 \times 10^6$ degrees/mm[34].

**Origin of the optical activity**. The optical activity is an effect which stems from non-locality[32]. As such, it can also stem from inhomogeneities in non-chiral structures, for example, from a transverse thickness gradient[75]. We have made all possible efforts to exclude the possibility that the optical activity observed in our experiments stems from this origin. For this, on the one hand, we have measured the thickness gradient of the structure experimentally, obtaining an average value of ≈1.3 nm/μm (see Supplementary Fig. 4). On the other hand, we have estimated the energy splitting which could stem from such gradient analytically and numerically, using full 3D electrodynamic simulations in COMSOL(R), and obtained no measureable splitting. To play any role in the effect, the gradient has to be at least 10 times higher.

As discussed in the main text, the OA arises in our sample at the scale of the whole structure, which becomes chiral because of the degeneracy of the modes of opposite parity. To confirm this statement, we perform a full numerical simulation with COMSOL(R), taking the parameters of the structure (perylene and mirror thickness) shown in Fig. 2. We did not include any microscopic optical activity in the model. In particular, we did not add the quartz layer, which could provide this activity (albeit small). We take the ordinary and extraordinary refractive indices of 1.7 and 2.5, respectively[52]. The excitonic effects are neglected. The results of the simulations are shown in Supplementary Fig. 5, showing the transmittivity as a function of in-plane wave vector $k_x$ and energy (similar to Fig. 2e). A clear anticrossing of the modes of opposite parity is observed at approximately the same energy and wave vector as in experiment. This confirms our interpretation of the effect in question.

## Data availability
The datasets generated during and/or analysed during the current study are available in the Open Science Framework (OSF) repository: https://osf.io/exhw2/?view_only=8000d5a60be1430986e6f24fa8c99cd1

## Code availability
The codes used for numerical simulations are available from the corresponding authors on reasonable request.

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

## Acknowledgements

We acknowledge useful discussions with M. Glazov. This work was supported by the National Key R&D Program of China (Grant No. 2018YFA0704805, 2018YFA0704802 and 2017YFA0204503), the National Natural Science Foundation of China (22090022, 21833005, 21873065, 21790364, 12074303, 11804267 and 21673144), the Beijing Natural Science Foundation of China (2192011), the High-level Teachers in Beijing Municipal Universities in the Period of 13th Five-year Plan (IDHT20180517 and CIT&TCD20180331), Beijing Talents Project (2019A23), the Open Fund of the State Key Laboratory of Integrated Optoelectronics (IOSKL2019KF01), Capacity Building for Sci-Tech Innovation-Fundamental Scientific Research Funds, Beijing Advanced Innovation Center for Imaging Theory and Technology. We acknowledge the support of the projects EU "QUANTOPOL" (846353), "Quantum Fluids of Light" (ANR-16-CE30-0021), of the ANR Labex Ganex (ANR-11-LABX-0014), and of the ANR program "Investissements d'Avenir" through the IDEX-ISITE initiative 16-IDEX-0001 (CAP 20-25). OB acknowledges support from the Australian Research Council Centre of Excellence in Future Low-Energy Electronics Technologies (CE170100039).

## Author contributions

J.R. – investigation, formal analysis, visualization, methodology, and writing; Q.L. – conceptualization, funding acquisition, methodology, resources, and supervision; F.L. – conceptualization, funding acquisition, methodology, supervision, writing, and project administration; Y.L. – methodology and writing; H.F. – conceptualization, funding acquisition, methodology, resources, and supervision; J.Y. – funding acquisition and supervision; D.S. – conceptualization, funding acquisition, formal analysis, methodology, visualization, and writing; G.M. – conceptualization, funding acquisition, methodology, writing, and supervision; O.B. – conceptualization, validation, methodology, visualization, and writing.

## Competing interests

The authors declare no competing interests.
