## [Peer Review File · Nature Communications]

Reviewers' Comments:

Reviewer #1:

Remarks to the Author:

In the manuscript "Nontrivial band geometry in an optically active system" the authors measured dispersion relation of perylene crystal embedded in a metallic-mirror microcavity. They observed an anti-crossing of photonic modes and interpreted these results as the presence of a giant optical activity (OA) in perylene. They proposed an effective Hamiltonian describing observed phenomena using some fitting parameters found from the experiment.

Although these results are interesting for the perspective of topological optics and the interest of scientific community in the field of spin-orbit interaction of light this paper needs to be improved since the main hypothesis – the role of OA (circular birefringence) – wasn't proved and all results could be explained in terms of birefringence.

There is a list of problems I found reading the article:

Problems:

1. Origin of giant OA of perylene.

The hypothesis of a giant OA due to the presence of perylene "excimers" should be clarified and justified, because such a giant optical activity was never observed in a pure perylene. Especially, OA of the order of 14000 deg/mm is typical for cholesteric liquid crystals (or "chiral stacks of 2D materials" pg. 20) but not for nonchiral symmetric molecules! The authors claim that the symmetry in perylene crystal is broken due to breaking of glide plane, which allows the OA. This effect potentially can lead to optical rotation, however such perturbation would result in an effect at least 2-3 orders of magnitudes weaker than reported in the manuscript. Moreover the authors cite Ref. 33 "Long-lying excited states in crystalline perylene" [Rangel et al. (2018) PNAS], where Rangel and co-workers showed that in the same crystallographic structure the direction of dipole moments (excitons) was oriented along the long molecular axis [Ref 33, Fig. 1D], and not perpendicularly to perylene "excimers" as proposed in the manuscript.

I understand that the role of the optical activity is the crucial point of this manuscript: it provides an approach different than a birefringence in <https://arxiv.org/abs/1912.09684> by A. Fieramosca, G. Malpuech, D. Solnyshkov et al. But I found no arguments that OA of such magnitude can be present in perylene crystal.

2. Interpretation of experimental data as the optical activity instead as the birefringence.

Alpha-perylene crystalizes in an "alpha" monoclinic phase of $p2_1/c$ symmetry [Pick A, et al. (2015) "Polymorph-selective preparation and structural characterization of perylene single crystals" *Cryst Growth Des* 15:5495–5504. ($a= 10.24 \text{ \AA}$, $b= 10.79 \text{ \AA}$, $c= 11.13 \text{ \AA}$, $\alpha=90 \text{ deg}$, $\beta= 100.92 \text{ deg}$, and $\gamma=90 \text{ deg}$)] and thus it is birefringent (probably it would be even biaxial). In supplementary materials to the publication of Pick (2015) there is Fig. S1 where the crystal structure of alpha-perylene is drawn. One can notice that a^* axis is tilted by 11 deg from the direction perpendicular to the (bc)-plane of perylene crystal. This suggests that also the optical axis of perylene crystal investigated in the manuscript wasn't perpendicular to the cavity plane. Such small tilt of the optical axis is sufficient to form "gapped Dirac cones" from "diabolic points" discussed in a similar paper <https://arxiv.org/abs/1912.09684> by A. Fieramosca, G. Malpuech, D. Solnyshkov et al. for a birefringent medium. This tilt can also explain asymmetries of Stokes parameters in Fig. 3.

The authors should carefully exclude the role of birefringence before claiming the leading role of optical activity. Small tilt of optical axis is sufficient to mix linearly polarized optical TE-TM modes. "The fact that the OA is linked with the exciton resonance is confirmed by the inversion of the sign of the effect in Fig. 1(B) of the main text" (pg. 19) can also be interpreted in terms of tilted optical axis.

3. Different (even-odd) parity of crossing photonic modes.

It was recently shown that photonic modes of different parities in a microcavity filled with birefringent medium can "interact" leading to "anticrossing" of eigenstates (Rechinska et al. "Photonic Engineering of Spin-Orbit Synthetic Hamiltonians in Liquid Crystal" *Science* (2019), 366, 727-730). According to this paper (supplementary inf.) the sign of TE-TM splitting depends on the difference of ordinary-extraordinary refractive indexes values (uniaxial, biaxial etc.). In Fig. 2E the authors marked E0 ("light mass" with minimum at 550nm) and E1 ("Heavy mass", 525 nm) eigenstates, which suggests that they have the same parity. However they can also mark E0 as "heavy" 600nm and E1 as "light" 550 nm, thus this pair can also have the same parity. In this manuscript I couldn't find arguments about the order of this splitting: is "light" above or below "heavy" eigenstate?

Due to the large TE-TM splitting in thick birefringent microcavity subsequent odd and even eigenstates can overlap at high k-vectors leading to "anticrossing" reported in the manuscript, which origin could be similar to what was observed by Rechinska et al. The authors should verify also this hypothesis. Again – small tilt of the optical axis mentioned in point 2 should favor anticrossing of modes.

4. Experimental results and their analysis

The authors measured "a polarization-resolved complete state tomography" (performed in all 6 polarizations). However they didn't show any results of such measurements – only the analysis was presented in Fig. 3. It would be useful for a reader to see all the data measured for Fig. 2DE and Fig. 3. (even as Supplementary Figures). What were the polarizations of the eigenstates observed in transmission measurement and how did they fit to the theoretical model? For instance: does the polarization of measured dispersion resemble supplementary Fig S1 AB? How the polarization of the transmission through perylene looks in k_x - k_y plane in the vicinity of "gapped Dirac cones"? Theoretical calculations should be compared with measurements of all Stokes vector components. In my opinion such comparison would help to exclude "OA approach" from "birefringence approach" or "odd-even approach".

5. Abelian gauge field

The authors claim the presence of non-Abelian gauge field. However TE-TM splitting and one linear term in momentum doesn't give an additional gauge field, see Shelykh (2018), *Phys. Rev. B* 98, 155428 "Optical analog of Rashba spin-orbit interaction in asymmetric polariton waveguides". The gauge field derived by authors is Abelian.

6. Figures.

In my opinion all figures don't fulfill standards of Nature Communications and should be modified. The authors use different units in different figures: angles or k-vectors, wavelength or energy, axes with or without ticks (Fig 1 DE), of different range (Fig 1 DE), etc. There should be somewhere definition of axes (x,y,z) in laboratory frame and corresponding axis directions of screw (if exist), birefringence, polarization of light etc.

Although this paper presents an interesting phenomena in my opinion the hypothesis of the leading role of the optical activity hasn't been proven yet. The theory is interesting, but the authors should find another system for investigations, where a large optical rotation will be present without doubt.

Therefore I don't recommend this article for publication in Nature Communications without serious changes in the text, which practically requires rewriting this article or doing the experiment again using chiral system.

Reviewer #2:

Remarks to the Author:

The manuscript presents the reconstruction of the Berry curvature and of the quantum metric in

an optically active material embedded in a microcavity. The strategy exploits a combination of effects on the polarization of light (TE-TM splitting, linear and circular birefringence), which allows the authors to model the response of system with a 2x2 model for the two light polarizations that displays topological features as a function of the incident wavevector.

The manuscript is well written and deserves publication once the comments and questions below are addressed.

1) The optical activity is described as a function of k_x and it is further discussed at the end of the Method section. However, it is not clear why there is no dependence on k_y . This fact should be properly commented and addressed.

2) In the introduction, the authors discuss the quantum anomalous Hall effect (QAHE) in photonics, where a Faraday effect (thus a magnetic field) is required. Immediately after, they move their discussion to the optical activity. The current presentation seems to suggest that optical activity is another way to obtain QAHE, which is not correct since it does not break time reversal symmetry (indeed the total Chern number of the Hamiltonian Eq.4 is zero). The authors should make more clear this distinction to the reader. Moreover, I would suggest to cite some relevant papers on the QAHE, as for example

Quantum Anomalous Hall Effect in $\text{Hg}_{1-y}\text{Mn}_y\text{Te}$ Quantum Wells
Chao-Xing Liu, Xiao-Liang Qi, Xi Dai, Zhong Fang, and Shou-Cheng Zhang
Phys. Rev. Lett. 101, 146802 – Published 1 October 2008

Experimental Observation of the Quantum Anomalous Hall Effect in a Magnetic Topological Insulator
Cui-Zu Chang et al.
Science 12 Apr 2013: Vol. 340, Issue 6129, pp. 167-170

- The mapping of Berry curvature and geometric tensor takes place through the Stokes vectors angles, as written in eq. 1 and more in detail in the theoretical paper Ref.37. The Stokes vectors act as a pseudospin on the Bloch sphere and parametrize the eigenvectors of the 2x2 model, thus allowing to extract the topological properties. However, it is not clear to me (and I could not clearly see the answer in Ref.37) how the experiment selects the eigenvector of a single band out of the two available. Indeed, eq. 1 should be band dependent. Could one see the result of the second band (which should have opposite response)?

- The authors mention a non-abelian gauge field in the abstract and in the introduction related to their results. However, no other reference to non-abelian gauge fields is made in the rest of the manuscript. To me it appears that all the results are for an abelian description instead. The author should clarify and eventually remove this claim.

- In Eq.5, the authors represent the "magnetic field in momentum space" that is typically used to represent 2x2 hamiltonians. I would suggest the authors to write the Hamiltonian as $H = \Omega \cdot \sigma$ in order to explain the notation, namely the meaning of Ω , which is currently only explained in words.

- It would be useful to plot the energy dispersion corresponding to the spectrum of hamiltonian in eq.4, assuming that the presented results concern a single band of the two.

- In Fig. 3D and 3E, the experimental results show a systematic squeezing of Berry curvature and quantum metric compared to the theory plot. Can the authors explain the origin of this phenomenon. Is some term missing in their modeling?

- Concerning the Hamiltonian in Eq.4, the authors could also comment on possible topological

transitions of this model or change in the topological features (e.g. the moving/merging of the Dirac points). Which parameters can be tuned in order to see any of these properties? Is it feasible?

- At pag.9 there should be a typo → “red \beta”

- Since the goal of this work is the reconstruction of topological and geometrical band properties (as the curvature or the metric), the authors should dedicate a part of the introduction to a description of the accomplishment already obtained in this direction in cold atoms and photonics, and they could take the opportunity to call the attention on what is different in their system and methodology compared to other methods or platforms. I provide below a list of some experimental and theoretical papers that the authors should cite, but the authors could also add other relevant ones.

Measurement of topological invariants in a 2D photonic system
Sunil Mittal, Sriram Ganeshan, Jingyun Fan, Abolhassan Vaezi & Mohammad Hafezi
Nature Photonics volume 10, pages180–183(2016)

Experimental measurement of the Berry curvature from anomalous transport
Martin Wimmer, Hannah M. Price, Iacopo Carusotto & Ulf Peschel
Nature Physics volume 13, pages545–550(2017)

Experimental reconstruction of the Berry curvature in a Floquet Bloch band
N. Fläschner, B. S. Rem, M. Tarnowski, D. Vogele, D.-S. Lühmann, K. Sengstock, C. Weitenberg
Science 27 May 2016, Vol. 352, Issue 6289, pp. 1091-1094

Mapping the Berry curvature from semiclassical dynamics in optical lattices
H. M. Price and N. R. Cooper
Phys. Rev. A 85, 033620 – Published 15 March 2012

Measuring the Chern number of Hofstadter bands with ultracold bosonic atoms
M. Aidelsburger, M. Lohse, C. Schweizer, M. Atala, J. T. Barreiro, S. Nascimbène, N. R. Cooper, I. Bloch & N. Goldman
Nature Physics volume 11, pages162–166(2015)

Measuring quantized circular dichroism in ultracold topological matter
Luca Asteria, Duc Thanh Tran, Tomoki Ozawa, Matthias Tarnowski, Benno S. Rem, Nick Fläschner, Klaus Sengstock, Nathan Goldman & Christof Weitenberg
Nature Physics volume 15, pages449–454(2019)

Tomography of Band Insulators from Quench Dynamics
Philipp Hauke, Maciej Lewenstein, and André Eckardt
Phys. Rev. Lett. 113, 045303 – Published 23 July 2014

Anomalous and Quantum Hall Effects in Lossy Photonic Lattices
Tomoki Ozawa and Iacopo Carusotto
Phys. Rev. Lett. 112, 133902 – Published 1 April 2014

Extracting the quantum metric tensor through periodic driving
Tomoki Ozawa and Nathan Goldman
Phys. Rev. B 97, 201117(R) – Published 31 May 2018

Response letter

In this document, we provide a point-by-point response to all comments of all reviewers. The changes to the manuscript are marked in red.

Reviewer #1 (Remarks to the Author):

Reviewer writes:

In the manuscript "Nontrivial band geometry in an optically active system" the authors measured dispersion relation of perylene crystal embedded in a metallic-mirror microcavity. They observed an anti-crossing of photonic modes and interpreted these results as the presence of a giant optical activity (OA) in perylene. They proposed an effective Hamiltonian describing observed phenomena using some fitting parameters found from the experiment.

Authors reply:

We thank the reviewer for providing an interpretation of our results, which clearly demonstrates that the reviewer has paid a particular attention to some aspects of our multidisciplinary work while overlooking others, which actually appear more central for us. Our manuscript is entitled "Nontrivial band geometry in an optically active system", and the central result of the manuscript is the measurement of this nontrivial geometry.

We would like to stress that this central result remains valid whatever the origin of optical activity of the system as a whole, and the presence of the latter is an undeniable experimental fact, evidenced by the circular polarization of the eigenmodes (as seen in Fig. 3(c), for example). However, we have excluded all possible origins of this activity, except the natural optical activity of the excitons in perylene, as we discuss in the manuscript and in our reply to the remarks below.

Reviewer writes:

Although these results are interesting for the perspective of topological optics and the interest of scientific community in the field of spin-orbit interaction of light this paper needs to be improved since the main hypothesis – the role of OA (circular birefringence) – wasn't proved and all results could be explained in terms of birefringence.

Authors reply:

We thank the reviewer for finding our results interesting for topological optics and spin-orbit interaction of light. We also thank the referee for pointing several alternative explanations, which for us were excluded from the start, but which are important to explicitly comment in the manuscript, which we do in the revised version.

Reviewer writes:

1. Origin of giant OA of perylene.

The hypothesis of a giant OA due to the presence of perylene "excimers" should be clarified and justified, because such a giant optical activity was never observed in a pure perylene. Especially, OA of the order of 14000 deg/mm is typical for cholesteric liquid crystals (or "chiral stacks of 2D materials" pg. 20) but not for nonchiral symmetric molecules! The authors claim that the symmetry in perylene crystal is broken due to breaking of glide plane, which allows the OA. This effect potentially can lead to

optical rotation, however such perturbation would result in an effect at least 2-3 orders of magnitudes weaker than reported in the manuscript. Moreover the authors cite Ref. 33 "Long-lying excited states in crystalline perylene" [Rangel et al. (2018) PNAS], where Rangel and co-workers showed that in the same crystallographic structure the direction of dipole moments (excitons) was oriented along the long molecular axis [Ref 33, Fig. 1D], and not perpendicularly to perylene

"excimers" as proposed in the manuscript.

Authors reply:

First, we never used the word "excimers", contrary to the statement of the reviewer, we were using the word "excitons". We have double checked the previously submitted file. Second, we do explain in the manuscript why the optical activity was never observed in a pure perylene: it is only present close to the exciton resonance and it is impossible to detect because of the linear birefringence. On the contrary, in a microcavity where this linear birefringence is compensated by the cavity TE-TM splitting, it is possible to observe a circular polarization of the modes due to the optical activity. Third, Fig. 1(d) of Ref. 33 is of course compatible with Fig. 3(a) of the same work: the dipole moment supposed by Fig. 3A has non-zero projections on the molecule plane and perpendicular to it, because the centers of the molecules within the pairs are shifted along their planes. It is also compatible with our Fig. S3, which indeed exhibits a non-zero overall projection along the molecule plane.

As to the amplitude of the effect, once the glide plane is broken, the system effectively becomes a chiral stack of 2D layers, with the exciton polarization changing between the layers, and therefore, contrary to the estimate of the referee, the effect can potentially be as large as in the stacks. We have added this comment to the corresponding part of the Methods section. It reads:

Indeed, once the glide plane is broken, the system effectively becomes a chiral stack of 2D layers, except that what is changing between the layers is not the orientation of the lattice vectors, but the orientation of the exciton polarization. Therefore the effect can potentially be as large as in the stacks.

Reviewer writes:

I understand that the role of the optical activity is the crucial point of this manuscript: it provides an approach different than a birefringence in <https://arxiv.org/abs/1912.09684> by A. Fieramosca, G. Malpuech, D. Solnyshkov et al. But I found no arguments that OA of such magnitude can be present in perylene crystal.

Authors reply:

Indeed, the optical activity, which manifests itself via the circular polarization of the eigenmodes at the anticrossing is a crucial difference with the arxiv by A. Fieramosca et al (where there is no anticrossing) and with Ref. 24 of the manuscript, where the anticrossing is produced by an applied magnetic field, breaking the TR symmetry.

Our arguments in favor of its presence in perylene crystal are given in the manuscript, they are also listed above. We also exclude all other explanations for the circular polarization of the optical modes, including the versions suggested by the referee (see below).

Reviewer writes:

2. Interpretation of experimental data as the optical activity instead as the birefringence.

*Alpha-perylene crystalizes in an "alpha" monoclinic phase of $p21/c$ symmetry [Pick A, et al. (2015) "Polymorph-selective preparation and structural characterization of perylene single crystals" *Cryst Growth Des* 15:5495–5504. ($a= 10.24$ Å, $b= 10.79$ Å, $c= 11.13$ Å, $\alpha=90$ deg, $\beta= 100.92$ deg, and $\gamma=90$ deg)] and thus it is birefringent (probably it would be even biaxial). In supplementary materials to the publication of Pick (2015) there is Fig. S1 where the crystal structure of alpha-perylene is drawn. One can notice that a^* axis is tilted by 11 deg from the direction perpendicular to the (bc) -plane of perylene crystal. This suggests that also the optical axis of perylene crystal investigated in the manuscript wasn't perpendicular to the cavity plane. Such small tilt of the optical axis is sufficient to form "gapped Dirac cones" from "diabolic points" discussed in a similar paper <https://arxiv.org/abs/1912.09684> by A. Fieramosca, G. Malpuech, D. Solnyshkov et al. for a*

birefringent medium. This tilt can also explain asymmetries of Stokes parameters in Fig. 3.

The authors should carefully exclude the role of birefringence before claiming the leading role of optical activity. Small tilt of optical axis is sufficient to mix linearly polarized optical TE-TM modes.

Authors reply:

We agree that a small tilt of the optical axis can mix the TE-TM optical modes for a given direction, and that if without the tilt that direction would correspond to a crossing of the branches, one would instead observe the anticrossing *for this particular direction*. However, we respectfully disagree that this could explain our experimental data. First of all, this tilt cannot explain the observation of any non-zero circular polarization degree (except in the very particular configuration of the point 3 of the referee, see below): the modes of the birefringent crystal are linear polarized, the TE-TM cavity modes are also linear polarized, and the modes of the global structure remain linear polarized, whatever the tilt of the optical axis. For a given direction in the reciprocal space, the tilt of the optical axis affects the modes of the structure and changes their polarization, for example, from horizontal to diagonal, but not to circular. As to the suggestion that this tilt can explain the observation of the "gapped Dirac cones": the tilt of the optical axis can explain that for a given direction the TE-TM and the birefringence do not compensate each other anymore. But there are necessarily two directions (different from the case with zero tilt) where the TE-TM and the linear birefringence do compensate each other. In other words, each diabolical point simply shifts in the reciprocal space. This is what the referee seems to suggest. However, since we performed the tomography of the whole 2D reciprocal space, we have actually obtained the dispersion of the modes (and their polarization, as shown in Fig. 3) in **all** directions of the reciprocal space, and not only for a single cross-section. These complete experimental results confirm that the anticrossing of the modes actually takes place. We comment on this in the text:

We note that Fig. 2(e) corresponds to the direction with the smallest gap, k_x . We have mapped the whole reciprocal space and we can therefore exclude the possibility that the observed anticrossing is simply due to a tilt of an optical axis which could shift the crossing point away.

Reviewer writes:

"The fact that the OA is linked with the exciton resonance is confirmed by the inversion of the sign of the effect in Fig. 1(B) of the main text" (pg. 19) can also be interpreted in terms of tilted optical axis.

Authors reply:

The change of the sign in Fig 1(b) is simply a confirmation that the effect is linked with the exciton resonance. As to the tilted optical axis, this suggestion does not explain the anticrossing of the modes and their circular polarization, and therefore is not sufficient to describe our data.

Reviewer writes:

3. Different (even-odd) parity of crossing photonic modes.

It was recently shown that photonic modes of different parities in a microcavity filled with birefringent medium can "interact" leading to "anticrossing" of eigenstates (Rechinska et al. "Photonic Engineering of Spin-Orbit Synthetic Hamiltonians in Liquid Crystal" Science (2019), 366, 727-730). According to this paper (supplementary inf.) the sign of TE-TM splitting depends on the difference of ordinary-extraordinary refractive indexes values (uniaxial, biaxial etc.). In Fig. 2E the authors marked E0 ("light mass" with minimum at 550nm) and E1 ("Heavy mass", 525 nm) eigenstates, which suggests that they have the same parity. However they can also mark E0 as "heavy" 600nm and E1 as "light" 550 nm, thus this pair can also have the same parity. In this manuscript I couldn't find arguments about the order of this splitting: is "light" above or below "heavy" eigenstate?

Due to the large TE-TM splitting in thick birefringent microcavity subsequent odd and even eigenstates can overlap at high k-vectors leading to "anticrossing" reported in the manuscript, which origin could be similar to what was observed by Rechinska et al. The authors should verify also this hypothesis. Again – small tilt of the optical axis mentioned in point 2 should favor anticrossing of modes.

Authors reply:

We thank the referee for this very good remark. Indeed, it is important to prove that our data cannot be interpreted as an anticrossing of the modes of different parity. This is particularly difficult for the sample studied in the manuscript, since, as correctly noted by the referee, the choice of the polarization doublets of the same mode number seems arbitrary.

However, the measurements performed on a thicker cavity, where a higher number of modes are observed at the same time (including two entire anticrossing doublets), confirm that these are the modes of the polarization doublet of the same mode number which exhibit the anticrossing. The figure shown below is now included in the manuscript as a Supplemental Figure 5.

The text added in the manuscript (in the Methods section) reads:

A very particular configuration is that of the crossing of the modes of opposite polarization and different parity [61]. In this case, chirality occurs not on the scale of individual molecules, but at the scale of the whole cavity. To exclude this interpretation of our data, we provide the results of additional measurements performed on a different sample, shown in Fig. S5. This sample is thicker, and its dispersion exhibits several pairs of modes of opposite polarization, at least three of which are visible in the figure. We mark two pairs of modes by their mode numbers n and $n+1$ and their polarization H and V (at $k=0$).

The polarization splitting (black arrows) at $k=0$ between the modes supposed to be of the same number ($E_{n,H}-E_{n,V}$) is smaller than in the sample studied in the main text, and the fact that the doublets are formed by the modes of the same number n or $n+1$ is confirmed by the fact that the inverse effective mass of the modes changes by the same factor as the polarization splitting $(E_{n+1,H}-E_{n+1,V})/(E_{n,H}-E_{n,V})\approx 2$, when approaching the exciton resonance. Indeed, the strong light-matter interaction in vicinity of the exciton resonance leads to the mixing of the excitonic and photonic modes. Closer to the resonance, the excitonic fraction of the modes increases, which increases their effective mass (excitons have a very large mass with respect to photons) and at the same time decreases the polarization splitting at $k=0$ (excitons exhibit no splitting, whereas photonic modes are split due to linear birefringence). The strong interaction of the excitonic and photonic resonances is also confirmed by the deviation of the observed modes from parabolicity visible in the figure, especially in panel a). Indeed, the observed dispersions exhibit a higher effective mass at higher wave vectors, whereas for purely photonic modes in a cavity the opposite would be expected.

The relation between the effective masses and the polarization splittings does not hold any more if one assumes that the modes are grouped differently. If the modes for which the anticrossing is observed were of the different parity (as in Ref. [61]), then the polarization splitting of the modes of the same parity would be given by the red arrows. Both of these correspond to approximately 200 meV without any notable difference. It is completely impossible for the polarization splitting to remain constant in spite of the strong overall change of the refractive index, as manifested by the change of the effective mass between the bands and along them. This hypothesis can therefore be rejected. We conclude that the optical activity that we observe does not arise from the anticrossing of the modes of different parity.

Reviewer writes:

4. Experimental results and their analysis

The authors measured "a polarization-resolved complete state tomography" (performed in all 6 polarizations). However they didn't show any results of such measurements – only the analysis was presented in Fig. 3. It would be useful for a reader to see all the data measured for Fig. 2DE and Fig. 3. (even as Supplementary Figures). What were the polarizations of the eigenstates observed in transmission measurement and how did they fit to the theoretical model? For instance: does the polarization of measured dispersion resemble supplementary Fig S1 AB? How the polarization of the transmission through perylene looks in k_x - k_y plane in the vicinity of "gapped Dirac cones"? Theoretical calculations should be compared with measurements of all Stokes vector components. In my opinion such comparison would help to exclude "OA approach" from "birefringence approach" or "odd-even approach".

Authors reply:

There is probably some misunderstanding. The results of the measurements *are* presented in Fig. 3(a,b,c), which show precisely the polarization of the eigenstates observed in the measurements. And indeed, they do resemble the supplementary Figure S1 (a,b), which can be seen from the color in Fig. 3(c). The polarization of the reflectivity (not the transmission, as the reviewer mentioned) in the whole k_x - k_y plane, including the vicinity of gapped Dirac cones, is again precisely what is shown in Fig 3(a,b,c). The caption of this figure reads "Measured Stokes parameters of the mode E0".

It is this measurement which allows to exclude the birefringence interpretation, whereas the odd-even hypothesis is excluded by the analysis of the alternative sample, also exhibiting the mode anticrossing.

We are sorry that the reviewer did not understand that the data she/he requested were already shown in the manuscript. We now explain the whole measurement and treatment procedure better in the Methods section, together with providing more experimental data. We also provide an explicit comparison of all 3 measured pseudospin components with the theoretical calculations. Because of this, we had to split the Figure 3 of the main text of the previous version into 2 figures: Fig. 3 and Fig. 4. Moreover, we have replaced the theoretical figure S1 by an experimental one, in order to stress the qualitative features of the behavior of the eigenstates at the anticrossing points and at the same time provide more experimental data.

The description of the two figures has been modified accordingly:

The validity of the effective Hamiltonian is confirmed by the measured 2D wave vector maps of the Stokes vector components of the lower branch, shown in Fig. 3(a-c) compared with theoretical predictions shown in panels (d-f). We note that the experimentally measured Stokes components are zero outside an elliptic region where the detection is efficient. Inside this region (marked with a white dashed line), the experiment and the theory exhibit a good agreement.

Reviewer writes:

5. Abelian gauge field

The authors claim the presence of non-Abelian gauge field. However TE-TM splitting and one linear term in momentum doesn't give an additional gauge field, see Shelykh (2018), Phys. Rev. B 98, 155428 "Optical analog of Rashba spin-orbit interaction in asymmetric polariton waveguides". The gauge field derived by authors is Abelian.

Authors reply:

We respectfully disagree with the statement of the referee: we do not claim that TE-TM together with a *linear* term give a gauge field. Our claim (based on Ref. 47 and several works in other systems, now cited in the text as new Refs. 44, 45) is that the TE-TM plus a *constant* term give rise to a gauge field in the vicinity of the crossing points (at non-zero k). Then, the optical activity close to these points can be considered as a constant Zeeman splitting (of the opposite signs for the two cones), which can be also incorporated into the non-Abelian Yang-Mills Hamiltonian (see Ref. 45). Our claim of the non-Abelian gauge field is therefore correct.

The added text reads:

In vicinity of the points $k_x = \pm k_0$, where the linear birefringence of the material is compensated by the TE-TM splitting, the Hamiltonian (4) can be written as a Rashba Hamiltonian with a constant Zeeman splitting, which was shown to be equivalent to a Hamiltonian of a non-relativistic quantum particle coupled to a non-Abelian Yang-Mills field [44,45].

However, in the outlook of the paper we consider the limiting case of small wave vectors (not in the vicinity of the anticrossing, but in the vicinity of $k=0$) of a fictitious system with zero birefringence. In this case, the system indeed cannot be considered as a non-Abelian gauge field (and we never claimed it), but it is rather an Abelian electromagnetic-like Hamiltonian with opposite magnetic fields for opposite spins (which is what we state). We now point this aspect more clearly in the text, when discussing Eq. (6). The new text reads:

We note that this field is Abelian, in spite of being spin-dependent.

Reviewer writes:

6. Figures.

In my opinion all figures don't fulfill standards of Nature Communications and should be modified. The authors use different units in different figures: angles or k-vectors, wavelength or energy, axes with or without ticks (Fig 1 DE), of different range (Fig 1 DE), etc. There should be somewhere definition of axes (x,y,z) in laboratory frame and corresponding axis directions of screw (if exist), birefringence, polarization of light etc.

Authors reply:

We have modified all figures according to the comments of the reviewer and to the standards of Nature Communications: all figures are now plotted with the same axes, showing wave vector and energy, which is the most qualitatively clear representation. As to the panels (d,e) (the reviewer is probably speaking of Fig. 2, not of Fig. 1) which had different scales, we note, however, that the two panels represent two different samples, and thus are not intended to be directly comparable. The panel d with the empty microcavity is simply provided to show the dispersion with the only ingredient being the TE-TM splitting.

Reviewer writes:

Although this paper presents an interesting phenomena in my opinion the hypothesis of the leading role of the optical activity hasn't been proven yet. The theory is interesting, but the authors should find another system for investigations, where a large optical rotation will be present without doubt.

Authors reply:

The central point of our work is the extraction of the quantum geometry of an optically active system, and this is accomplished beyond any doubt, since the optical activity of the whole system (the circular polarization of the modes) is explicitly measured experimentally and has not been doubted even by the reviewer. We hope that with the new arguments provided in the reply and in the text, the reviewer will agree that since no other explanations are possible, the origin of the optical activity that we suggest must be the correct one.

Reviewer writes:

Therefore I don't recommend this article for publication in Nature Communications without serious changes in the text, which practically requires rewriting this article or doing the experiment again using chiral system.

Authors reply:

We hope that now that we have pointed out the misunderstandings of the reviewer (concerning the presence of experimental results in the manuscript) and provided more details, experimental data, and theoretical arguments, the reviewer will recommend our manuscript for publication in Nature Communications.

Reviewer #2 (Remarks to the Author):

Reviewer writes:

The manuscript presents the reconstruction of the Berry curvature and of the quantum metric in an optically active material embedded in a microcavity. The strategy exploits a combination of effects on the polarization of light (TE-TM splitting, linear and circular birefringence), which allows the authors to model the response of system with a 2x2 model for the two light polarizations that displays topological features as a function of the incident wavevector.

Authors reply:

We thank the reviewer for correctly summarizing our results.

Reviewer writes:

The manuscript is well written and deserves publication once the comments and questions below are addressed.

Authors reply:

We thank the reviewer for the positive appreciation. We have replied all comments and questions and corrected the manuscript accordingly. See details below.

Reviewer writes:

1) The optical activity is described as a function of k_x and it is further discussed at the end of the Method section. However, it is not clear why there is no dependence on k_y . This fact should be properly commented and addressed.

Authors reply:

We thank the reviewer for this remark. For symmetry reasons, the optical activity must change sign at $k=0$ (see Fig 1B, top part). Therefore, there is necessarily a line in the reciprocal space along which the corresponding term is zero. Because of the symmetry of the crystal structure, this line also has to be parallel to one of the two axes of the linear birefringence. This is why it is zero along k_y . The new text added in the Methods section in order to comment on this reads:

For symmetry reasons, the optical activity must change sign at $k=0$ (see Fig 1(b), top part). Therefore, there is necessarily a line in the reciprocal space, passing through $k=0$, along which the OA is zero. Because of the symmetry of the crystal structure, this line also has to be parallel to one of the two axes of the linear birefringence. This is why the leading OA term is linear in k_x and there is no dependence on k_y . This term can be deduced as follows.

Reviewer writes:

2) In the introduction, the authors discuss the quantum anomalous Hall effect (QAHE) in photonics, where a Faraday effect (thus a magnetic field) is required. Immediately after, they move their discussion to the optical activity. The current presentation seems to suggest that optical activity is another way to obtain QAHE, which is not correct since it does not break time reversal symmetry (indeed the total Chern number of the Hamiltonian Eq.4 is zero). The authors should make more clear this distinction to the reader. Moreover, I would suggest to cite some relevant papers on the QAHE, as for example

Authors reply:

We completely agree that the optical activity does not allow to obtain QAHE, and we now stress this clearly in the text (page 4). We have added both references suggested by the referee to the manuscript. The new text reads:

Of course, QAHE is not limited to photonics: it has been originally proposed [12] and recently demonstrated in electronics [13,14] and atomic lattices [15].

Reviewer writes:

- The mapping of Berry curvature and geometric tensor takes place through the Stokes vectors angles, as written in eq. 1 and more in detail in the theoretical paper Ref.37. The Stokes vectors act as a pseudospin on the Bloch sphere and parametrize the eigenvectors of the 2x2 model, thus allowing to extract the topological properties. However, it is not clear to me (and I could not clearly see the answer in Ref.37) how the experiment selects the eigenvector of a single band out of the two available. Indeed, eq. 1 should be band dependent. Could one see the result of the second band (which should have opposite response)?

Authors reply:

The reviewer is completely right, the polarization, and therefore the Berry curvature, of the second band is opposite with respect to the first one. We now discuss the selection of the eigenstates more in details in the Methods section. We provide more experimental data as well, as requested also by Reviewer 1. In particular, we have replaced the theoretical Supplemental figure 1 (showing dispersion and pseudospin) by an experimental figure, where, for one direction in the reciprocal space passing through the anticrossing point, the extracted dispersion is clearly visible, together with the orientation of the extracted pseudospin, which is indeed opposite for the two bands. The new text reads:

We consider a reflectivity spectrum measured under white light excitation for total intensity, such as shown in Fig. S7 (black circles). We first determine the wavelength λ_0 and the energy E_0 corresponding to the particular mode, by fitting the total reflection spectrum with Lorentzian-broadened resonances over an approximately linear background (red solid line). We then fit the individual intensity components to determine the relative weight of resonance (taking into account the magnitude and the width of the peak) in each of the 6 polarizations (blue and cyan triangles in Fig. S7 for experimental circular polarization and red dashed and dash-dotted lines for theory), which allows finally to determine the 3 components of the Stokes vector. In our example, we show only two polarization projections of the six (to avoid overloading the figure). We note that the positions of the reflectivity minima detected in two polarizations under a non-polarized excitation do not necessarily correspond to the positions of the modes: their position depends on the linewidth and the polarization degree, and the maximal deviation can be of the order of the linewidth. As an example, the reflectance in the two circular polarizations is given by:

Reviewer writes:

- The authors mention a non-abelian gauge field in the abstract and in the introduction related to their results. However, no other reference to non-abelian gauge fields is made in the rest of the manuscript. To me it appears that all the results are for an abelian description instead. The author should clarify and eventually remove this claim.

Authors reply:

We now discuss this point better, adding several references, as requested also by Reviewer 1 (see above for the inserted text appearing in pages 10 and 17). The essential idea is that the Hamiltonian (4) in vicinity of an anticrossing or a crossing can be considered as a Hamiltonian of a massive particle in a non-Abelian Yang-Mills field. This is not the central point of the manuscript, but it is worth being mentioned, since it is one of the first observations of such configurations in photonics.

Reviewer writes:

- In Eq.5, the authors represent the “magnetic field in momentum space” that is typically used to represent 2x2 hamiltonians. I would suggest the authors to write the Hamiltonian as $H = \Omega \cdot \sigma$ in order to explain the notation, namely the meaning of Ω , which is currently only explained in words.

Authors reply:

We thank the reviewer for this remark. We have now provided an explicit reformulation of the Hamiltonian according to the advice of the reviewer. The new text reads:

it is a linear combination of Pauli matrices that can be physically interpreted as an effective magnetic field acting on the Stokes vector, that is, $H_k = \Omega \sigma$, where σ is a vector of Pauli matrices and Ω is the effective field

Reviewer writes:

- It would be useful to plot the energy dispersion corresponding to the spectrum of hamiltonian in eq.4, assuming that the presented results concern a single band of the two.

Authors reply:

This dispersion is shown in Fig 2(e) by dashed lines. We now stress it more clearly in the text (this was only written in the figure caption). The figure has been replotted as $E(k)$, so that it really corresponds directly to the Hamiltonian (4).

Reviewer writes:

- In Fig. 3D and 3E, the experimental results show a systematic squeezing of Berry curvature and quantum metric compared to the theory plot. Can the authors explain the origin of this phenomenon. Is some term missing in their modeling?

Authors reply:

We thank the reviewer for this interesting remark. The reason of this squeezing might be the non-parabolicity of the dispersion in vicinity of the exciton resonance, which is indeed neglected in the simple 2x2 Hamiltonian, but also the precision of the measurements. We now comment on this in the text.

The new text reads:

The distortion of the maxima of the Berry curvature in experiment might be explained by the non-parabolicity of the dispersion in vicinity of the exciton resonance and by the difference of the experimental resolution in the two directions.

Reviewer writes:

- Concerning the Hamiltonian in Eq.4, the authors could also comment on possible topological transitions of this model or change in the topological features (e.g. the moving/merging of the Dirac points). Which parameters can be tuned in order to see any of these properties? Is it feasible?

Authors reply:

We thank the reviewer for this interesting remark! Although no topological transitions in the strict sense are possible because the Chern number of each band always remains zero, the band geometry can indeed be strongly modified by tuning the effective Hamiltonian parameters. We now discuss it better in the text. The new text reads:

Tuning the coefficients of the Hamiltonian by choosing different materials would allow to deeply modify the band geometry. In Ref. [24], a system described by a similar effective Hamiltonian was studied, except that the OA was replaced by an effective Zeeman splitting. The bands were showing two split Dirac cones, like in the present work, but with the same sign for the Berry curvature in a given band. By tuning the linear birefringence β_0 to zero in Eq. (4), the dispersion bands would exhibit a crossing at $k=0$, but the reciprocal space nevertheless remains split into two "valleys" of opposite Berry curvature because of the OA. Instead of being concentrated at the anticrossing points, the Berry curvature exhibits a crescent shape in this case. If both OA and birefringence are set to zero, which is the case of an empty cavity shown in Fig. 2(d), the dispersion represents two touching parabolas, which are characterized by two Berry monopoles of opposite charges at $k=0$.

Reviewer writes:

- At pag.9 there should be a typo → "red \beta"

Authors reply:

We thank the reviewer! Typo corrected.

Reviewer writes:

- Since the goal of this work is the reconstruction of topological and geometrical band properties (as the curvature or the metric), the authors should dedicate a part of the introduction to a description of

the accomplishment already obtained in this direction in cold atoms and photonics, and they could take the opportunity to call the attention on what is different in their system and methodology compared to other methods or platforms. I provide below a list of some experimental and theoretical papers that the authors should cite, but the authors could also add other relevant ones.

Authors reply:

We thank the reviewer for these relevant references. We have extended the discussion, adding all of these references, but indeed also some other ones, which have appeared just recently (such as Ref. 60). The new discussion reads:

Quantum geometry is currently a subject of active studies. It is studied both globally, at the level of topological invariants, and locally, as a distribution of the Berry curvature and quantum metric in a certain parameter space. Topological invariants with their discrete integer values and associated global effects, such as the presence of edge states determined via the bulk-edge correspondence, are easier to be measured experimentally [50-53]. The local distribution of the Berry curvature and, later, the quantum metric, have usually been measured via the related dynamical effects [54-60], such as the anomalous Hall drift. Recently, the quantum geometry has been extracted from the eigenstates of a photonic system, like in the present work, with an additional confirmation of the results by the anomalous Hall measurements [24].

We thank both reviewers once again for their constructive criticism, which has allowed us to improve our work. We hope that the reviewers will accept the revised version of the manuscript for publication in Nature Communications.

Sincerely yours,

The authors

Reviewers' Comments:

Reviewer #1:

Remarks to the Author:

I would like to thank Authors for all the corrections and comments. The manuscript presents an interesting physics and in my opinion it is worth publishing in Nature Communications. What's more, I suspect that the authors are not aware of the discovery they have made and insist on misinterpreting the phenomenon.

I agree with the Authors that "that the central result of the manuscript is the measurement of this nontrivial geometry". However, in addition to the description of the interesting physical phenomenon itself, it is important to provide its true origin. Nature Communications is a respectable journal because it is well known for publishing high quality research and one cannot disseminate unreliable data that conflicts with existing scientific findings and theoretical reasoning.

1. Perylene, according to current knowledge, does not have optical activity (OA). It means that (paraphrasing the Authors statement in the Methods) "chirality in perylene does not occur on the scale of individual molecules".

2. The results of the experiment can be described by birefringence of crystalline perylene embedded in an optical cavity. It means that "chirality occurs at the scale of the whole cavity" as in [61].

Ad 1.

Exciton polarization cannot be responsible for a microscopic "chiral stack" leading to OA. Otherwise, would this hypothetical "excitonic polarization" be permanent? Apart from the fact that wavelengths greater than 470 nm are not practically absorbed by perylene, estimate the number of photons required to keep excitons present in this system to form a "stack of exciton polarization", etc. Do the authors suggest that when they stop illuminating the sample with violet light (and stop creating excitons) the whole effect of OA will disappear? The Authors wrote on pg. 4 of the manuscript "Perylene (C₂₀H₁₂) under its crystallized form is a specific organic microcrystal showing strong OA due to its crystal structure" without providing any references to OA of perylene. They cannot find any, because this statement is incorrect. Finding new mechanism of exciton-polarization-induced strong OA of a non-chiral perylene molecule in p21/c symmetry crystalline structure (probably the most frequently occurring space group) could be of the major discovery of this paper, more important than the presence of Berry phase!

Ad 2.

In the "Response letter on pg 2" the Authors concluded "this central result (of this manuscript) remains valid whatever the origin of optical activity of the system as a whole, and the presence of the latter is an undeniable experimental fact, evidenced by the circular polarization of the eigenmodes (as seen in Fig. 3(c), for example)." If the theory is correct, then what is the origin of observed OA and anticrossings of optical modes?

I'm sending three figures showing results of numerical modeling of the perylene crystal reflectivity observed in linear (horizontal-vertical) and circular (left and right) polarization together with DOCP (S3) and DOLP (S1) (Degree of Circular / Linear Polarization). I estimated the energies of these modes in Fig. S5b and I used material parameters found in the literature (the details of my calculations are in "Methods" below). I also attached numerical results for ($n \cdot d = 2200$ nm, anticrossings), ($n \cdot d = 2000$ nm, anticrossings) and ($n \cdot d = 3000$ nm, intersections).

One can easily see anticrossings, which originates from large birefringence of perylene and the resonance of modes with subsequent numbers n and $n+1$.

My inspiration has come from the discussion of Ref. Rechcinska [61] included in the "Response letter" and a new version of the manuscript. In fact, thanks to the use of an optical cavity, the authors of the manuscript managed to artificially induce OA in a birefringent material. When some

conditions are fulfilled – i.e. linearly polarized optical modes with subsequent numbers n and $n+1$ are in the resonance at certain k -vector, similarly to Ref. [61] - the electromagnetic wave passing through the cavity is circularly polarized in two opposite directions - and in these directions a non-zero Berry curvature is observed. The Rashba Hamiltonian (4) proposed by Authors describes an approximate solution around these interesting points. In their experiment the Authors cannot tilt the optical axis as in [61], but they profit from unique properties of the perylene: large optical anisotropy (resulting with linearly polarized modes of very different photon effective masses) and crystalline structure (with well-defined direction of optical axis). Some luck in the experiment was also needed as usual, because the thickness of perylene microcrystal was also important, e.g. I modeled $n*d=3000$ nm the optical modes accessible in the experiment (± 70 deg) only intersect.

Thus I would suggest the Authors to change the part of the manuscript related to the OA of perylene (what fundamentally cannot be observed "on the scale of individual molecules") and to explain this new method for coupling of optical modes and thus formation of non-trivial band geometry.

"Methods"

I devoted some time to expand the transfer matrix method with the Berreman 4x4 approach, which works well in the description of birefringent systems.

1. Alpha-perylene crystalizes in an alpha monoclinic phase of $p2_1 / c$ symmetry. [Pick A, et al. (2015) "Polymorph-selective preparation and structural characterization of perylene single crystals" *Cryst Growth Des* 15: 5495-5504. ($a = 10.24$ Å, $b = 10.79$ Å, $c = 11.13$ Å, $\alpha = 90$ deg, $\beta = 100.92$ deg, and $\gamma = 90$ deg)]. The optical axis of alpha-perylene is tilted about 11 degrees from the crystal plane on Fig.S2a.
2. From Fig. S5b, knowing the order of optical modes, one can estimate the thickness of the cavity d - the product of the refractive index $n * d = 2200$ nm.
3. Assuming ordinary and extraordinary refractive indexes of perylene about 1.7 and 2.5 it is possible to obtain anticrossings. The perylene refractive indexes can be found in literature [e.g. "Low-lying excited states in crystalline perylene" Rangel et al *Proc Natl Acad Sci US A*. 2018 Jan 9; 115 (2): 284–289, doi: 10.1073 / pnas.1711126115]. However they can be independently estimated from the curvature of the optical modes in x and y polarization. One can use negative or positive birefringence, the results are practically the same.
4. I attach two zip files: the first one with 3 figures of reflectivity, angle (-70 deg, 70 deg), wavelength (440 nm, 620 nm) perylene thickness ($n*d=2000$ nm, 2200 nm and 3000 nm) and the second zip file with numerical data of these images.

Reviewer #2:

Remarks to the Author:

I have read the new version of the manuscript where the authors have taken into account my comments. I still have a couple of minor remarks that I think should be addressed, but besides those I am in favour of publication of the manuscript.

1) In order to use the wording non-Abelian, the authors have to explore the non-commutativity of the gauge field, which is indeed measured in Soljagic's paper that they are now citing. If this is not the case here, such a claim of realization of a non-Abelian gauge field is not correct, in my opinion. Indeed, non-Abelian features emerge when observing the effect of off-diagonal components of the Berry connection, of the Berry curvature or the quantum metric. Since the authors' results are

for single bands or single modes, namely only the diagonal terms of these tensors, their results have no evidence of non-Abelian gauge fields. In particular, when the authors write

“We provide a direct measurement of the complete band geometry, namely the Berry curvature and the quantum metric of the photonic bands. ”: The word complete seems to suggest that all the components of the tensors are measured, namely also the off-diagonal ones.

“We show that their non-trivial distribution can be interpreted as the result of the action of a non-Abelian gauge field”: As before it seems that you have a clear evidence of the non-Abelian character that allows such interpretation, which is not possible if you only measure the diagonal terms of the tensors. The model may have non-Abelian properties but not the measurements.

In conclusion, it is correct when the authors say “the Hamiltonian (4) can be written as a Rashba Hamiltonian with a constant Zeeman splitting, which was shown to be equivalent to a Hamiltonian of a non-relativistic quantum particle coupled to a non-Abelian Yang-Mills field” but the previous sentences are not. In other words, the authors have an effective model possessing non-Abelian properties but those are not shown in the manuscript in any part.

Besides fixing the two sentences cited above, I would then remove the wording non-Abelian from the abstract as well to avoid misleading the readers. In particular, they write “Photonic spin-orbit-coupling induced by the cavity results in the action of a non-Abelian gauge field on photons.”, which is not correct in my opinion for the reasons given above but also because it makes the reader think that this is a real-space gauge field without further clarification.

2) Is the parabolicity really a possible explanation? Can the authors provide evidence of this or show how that occurs by modifying the Hamiltonian accordingly? Can the such modeling be benchmarked and tested with the experimental data? I think this is a minor concern but it is important to provide a clear evidence of such claims and a correct interpretation of the data, if possible, whereas the current statement seems yet unsupported.

Response letter

In this document, we provide a point-by-point response to all comments of all reviewers. The changes to the manuscript are marked in red.

Reviewer #1 (Remarks to the Author):

Reviewer writes:

I would like to thank Authors for all the corrections and comments. The manuscript presents an interesting physics and in my opinion it is worth publishing in Nature Communications. What's more, I suspect that the authors are not aware of the discovery they have made and insist on misinterpreting the phenomenon.

Authors reply: We thank the reviewer for the positive appreciation of our work and for the persistence in helping us to choose the correct interpretation of the phenomenon. The arguments given by the reviewer below have finally convinced us that the interpretation supported by the referee is the correct one. We have therefore rewritten the whole manuscript in light of this interpretation.

Reviewer writes:

I agree with the Authors that "that the central result of the manuscript is the measurement of this nontrivial geometry". However, in addition to the description of the interesting physical phenomenon itself, it is important to provide its true origin. Nature Communications is a respectable journal because it is well known for publishing high quality research and one cannot disseminate unreliable data that conflicts with existing scientific findings and theoretical reasoning.

Authors reply: We agree with the reviewer and we express our gratitude once again for helping us to choose the correct interpretation.

Reviewer writes:

Exciton polarization cannot be responsible for a microscopic "chiral stack" leading to OA. Otherwise, would this hypothetical "excitonic polarization" be permanent? Apart from the fact that wavelengths greater than 470 nm are not practically absorbed by perylene, estimate the number of photons required to keep excitons present in this system to form a "stack of exciton polarization", etc. Do the authors suggest that when they stop illuminating the sample with violet light (and stop creating excitons) the whole effect of OA will disappear? The Authors wrote on pg. 4 of the manuscript "Perylene (C₂₀H₁₂) under its crystallized form is a specific organic microcrystal showing strong OA due to its crystal structure" without providing any references to OA of perylene. They cannot find any, because this statement is incorrect. Finding new mechanism of exciton-polarization-induced strong OA of a non-chiral perylene molecule in $p21/c$ symmetry crystalline structure (probably the most frequently occurring space group) could be of the major discovery of this paper, more important than the presence of Berry phase!

Authors reply: This argument of the reviewer has been the most convincing for us. We understood that we were wrong, because the exciton-related symmetry-breaking effects must necessarily be nonlinear and require polarized pumping. There are such experiments indeed, working best in the pump-probe configuration, but in our case the pumping is unpolarized and the whole set of measurements is linear. Therefore, the interpretation we had suggested is wrong. We thank the referee for pointing it out.

Reviewer writes:

In fact, thanks to the use of an optical cavity, the authors of the manuscript managed to artificially induce OA in a birefringent material. When some conditions are fulfilled – i.e. linearly polarized optical modes with subsequent numbers n and $n+1$ are in the resonance at certain k -vector, similarly to Ref. [61] - the electromagnetic wave passing through the cavity is circularly polarized in two opposite directions - and in these directions a non-zero Berry curvature is observed. The Rashba Hamiltonian (4) proposed by Authors describes an approximate solution around these interesting points. In their experiment the Authors cannot tilt the optical axis as in [61], but they profit from unique properties of the perylene: large optical anisotropy (resulting with linearly polarized modes of very different photon effective masses) and crystalline structure (with well-defined direction of optical axis).

Thus I would suggest the Authors to change the part of the manuscript related to the OA of perylene (what fundamentally cannot be observed "on the scale of individual molecules") and to explain this new method for coupling of optical modes and thus formation of non-trivial band geometry.

Authors reply:

We have followed the advice of the reviewer and rewritten the manuscript as suggested. The changes concern the origin of emergent optical activity at the scale of the cavity, with the mechanism explained in the revised Fig. 1. We warmly thank the reviewer for the efforts devoted to the simulation of the polarization-resolved reflectivity of the cavity with perylene. Our own simulations based on COMSOL are now shown in the revised Supplemental Materials (Figure S5) and commented in the Methods section.

Reviewer #2 (Remarks to the Author):

Reviewer writes:

I have read the new version of the manuscript where the authors have taken into account my comments. I still have a couple of minor remarks that I think should be addressed, but besides those I am in favour of publication of the manuscript.

Authors reply:

We thank the reviewer for the positive appreciation of our work and for the recommendation to publish the manuscript. We have revised the manuscript as suggested by the reviewer.

Reviewer writes:

1) In order to use the wording non-Abelian, the authors have to explore the non-commutativity of the gauge field, which is indeed measured in Soljagic's paper that they are now citing. If this is not the case here, such a claim of realization of a non-Abelian gauge field is not correct, in my opinion. Indeed, non-Abelian features emerge when observing the effect of off-diagonal components of the Berry connection, of the Berry curvature or the quantum metric. Since the authors' results are for single bands or single modes, namely only the diagonal terms of these tensors, their results have no evidence of non-Abelian gauge fields. In particular, when the authors write

"We provide a direct measurement of the complete band geometry, namely the Berry curvature and the quantum metric of the photonic bands. ": The word complete seems to suggest that all the components of the tensors are measured, namely also the off-diagonal ones.

“We show that their non-trivial distribution can be interpreted as the result of the action of a non-Abelian gauge field”: As before it seems that you have a clear evidence of the non-Abelian character that allows such interpretation, which is not possible if you only measure the diagonal terms of the tensors. The model may have non-Abelian properties but not the measurements.

In conclusion, it is correct when the authors say “the Hamiltonian (4) can be written as a Rashba Hamiltonian with a constant Zeeman splitting, which was shown to be equivalent to a Hamiltonian of a non-relativistic quantum particle coupled to a non-Abelian Yang-Mills field” but the previous sentences are not. In other words, the authors have an effective model possessing non-Abelian properties but those are not shown in the manuscript in any part.

Authors reply: We agree with the reviewer. The effective field is non-Abelian, but we do not carry out any measurements capable of showing its non-Abelian nature (in this particular work). We have revised the text accordingly, as suggested below by the reviewer.

Reviewer writes:

Besides fixing the two sentences cited above, I would then remove the wording non-Abelian from the abstract as well to avoid misleading the readers. In particular, they write “Photonic spin-orbit-coupling induced by the cavity results in the action of a non-Abelian gauge field on photons.”, which is not correct in my opinion for the reasons given above but also because it makes the reader think that this is a real-space gauge field without further clarification.

Authors reply: We have removed the wording “non-Abelian” from the abstract and from the introduction of the manuscript, following the advice of the reviewer. We have kept the references to the works where the Rashba Hamiltonian has been interpreted as a non-Abelian gauge field and where the consequences of the non-Abelian nature of this field have been evidenced.

Reviewer writes:

2) Is the parabolicity really a possible explanation? Can the authors provide evidence of this or show how that occurs by modifying the Hamiltonian accordingly? Can the such modeling be benchmarked and tested with the experimental data? I think this is a minor concern but it is important to provide a clear evidence of such claims and a correct interpretation of the data, if possible, whereas the current statement seems yet unsupported.

Authors reply: We have now stated our idea for this possible explanation more explicitly in the manuscript and added a small section into Methods and an extra Supplementary Figure (S3). We discuss the physical effect which is at the origin of non-parabolicity: the strong coupling with excitons. With a strongly-coupled 4x4 Hamiltonian (taking a value of the Rabi splitting obtained in Ref. 46), we calculate the Berry curvature distribution and demonstrate that it exhibits maxima of the same shape as in experiment. However, since all physics is already captured well by a 2x2 effective Hamiltonian and since the strong coupling is not the main topic of our work, we keep the simplest possible description throughout the manuscript.

We hope that both reviewers will find the revised manuscript suitable for publication in Nature Communications.

Sincerely yours,

The authors

Reviewers' Comments:

Reviewer #1:

Remarks to the Author:

In my opinion this manuscript has been greatly improved. The Authors have independently tested my suppositions about the origin of the optical activity using their own COMSOL simulations and they have obtained perfect agreement with the experiment. They adopted these new results and applied their theoretical reasoning using the concept of the "emergent OA". I'm very glad that the review process has converged resulting in a very good paper. And I would like to thank the Authors for an acknowledgement of my modest support.

Some minor remarks (I would like to leave the Authors the decision if they would like to apply them. They are so minor, that I do not need to read this manuscript again before the acceptance by the Editor).

The authors use linearly polarized TE-TM modes and H-V modes. At a given laboratory frame indeed TE=H and TE=V (Fig. 1b, Poincare sphere), but in general one should be careful. In my opinion linearly polarised and split H-V modes in birefringent crystals mean that we have no longer "TE-TM splitting". For instance, when we rotate the sample then H-V polarisation also rotates (e.g. split H mode becomes V when sample is rotated by 90 deg because the optical axis has been also rotated), while TE-TM stay the same (because it depends on the plane of incidence). Additionally, H-V splitting is also k-dependent because of the difference in effective masses. Therefore on pg. 12 in the sentence "As expected, the linear birefringence is compensated by the k-dependent TE-TM field at the anticrossing points" I would rather use "k-dependent H-V" etc. But I might be wrong. I would like to ask the Authors to verify carefully if in all paragraphs (and in Fig. 1b) the meaning of TE-TM and H-V modes has not been mixed up.

Another minor remark: Fig S5 is in nm while all others figures are in eV. This is why the parabola has a "negative curvature". I would advise to plot all figures, also in Suppl. Mater., in energy, not in wavelengths.

Jacek Szczytko.

Reviewer #2:

Remarks to the Author:

I have revised the new version of the manuscript. All my comments have been addressed and I therefore recommend publication in its current form.

Reviewer #3:

None

Point-by-point response

In this document, we provide a point-by-point response to the comments of the reviewers.

Response to the Reviewer #1

Reviewer writes:

In my opinion this manuscript has been greatly improved. The Authors have independently tested my suppositions about the origin of the optical activity using their own COMSOL simulations and they have obtained perfect agreement with the experiment. They adopted these new results and applied their theoretical reasoning using the concept of the "emergent OA". I'm very glad that the review process has converged resulting in a very good paper. And I would like to thank the Authors for an acknowledgement of my modest support.

Authors respond:

We thank the reviewer once again. We have extended the acknowledgement to the reviewer 1, which now reads:

"We also acknowledge the exceptional contribution of the reviewer 1, who has suggested the correct origin of the optical activity and managed to convince us."

Reviewer writes:

Some minor remarks (I would like to leave the Authors the decision if they would like to apply them. They are so minor, that I do not need to read this manuscript again before the acceptance by the Editor).

Authors respond:

We thank the reviewer for these remarks, to which we reply below.

Reviewer writes:

The authors use linearly polarized TE-TM modes and H-V modes. At a given laboratory frame indeed $TE=H$ and $TE=V$ (Fig. 1b, Poincare sphere), but in general one should be careful. In my opinion linearly polarised and split H-V modes in birefringent crystals mean that we have no longer "TE-TM splitting". For instance, when we rotate the sample then H-V polarisation also rotates (e.g. split H mode becomes V when sample is rotated by 90 deg because the optical axis has been also rotated), while TE-TM stay the same (because it depends on the plane of incidence). Additionally, H-V splitting is also k-dependent because of the difference in effective masses. Therefore on pg. 12 in the sentence "As expected, the linear birefringence is compensated by the k-dependent TE-TM field at the anticrossing points" I would rather use "k-dependent H-V" etc. But I might be wrong. I would like to ask the Authors to verify carefully if in all paragraphs (and in Fig. 1b) the meaning of TE-TM and H-V modes has not been mixed up.

Authors respond:

The reviewer has correctly pointed out that the total splitting between the linear polarizations of the eigenmodes and their resulting orientation are not aligned any more with the TE and TM orientations. This is already correctly taken into account in our model, by the two terms β_0 (birefringence or HV splitting) and β (TE-TM splitting) in the effective Hamiltonian (5). We have checked that our notations are consistent: the splitting associated with the birefringence is called "HV" and stays "HV"

precisely because the sample is not rotated. The splitting associated with the plane of incidence is called TE-TM and its eigenstates change depending on the propagation direction. We suppose that the question of the reviewer has appeared because of the impression that we rotate the sample, while in fact the sample is not rotated, whereas the plane of incidence does rotate. We now stress it explicitly in the text while discussing the two contributions after Eq. (3):

The important difference between the TE-TM and the H-V splittings is that the orientation of TE-TM is defined by the in-plane wave vector (the orientation of the incidence plane with respect to the polarization-resolving detector), while the orientation of the H-V splitting is linked with the crystal axes and thus stays constant, because the sample is not rotated in our experiments.

We have also replaced “our case” by “this case” in the description of figure 1, to avoid the false impression that we use HV and TE-TM as synonyms for all figures, whereas they just coincide for this particular figure associated with a particular propagation direction. The phrase now reads:

But the reflection on a metallic mirror also leads to the polarization inversion with respect to a different axis (TE-TM or HV in **this** case, also marked in Fig. 1(b))

Reviewer writes:

Another minor remark: Fig S5 is in nm while all others figures are in eV. This is why the parabola has a "negative curvature". I would advise to plot all figures, also in Suppl. Mater., in energy, not in wavelengths.

Authors respond:

We have replotted Fig. S5 in eV, as the other figures. We thank the reviewer for this advice, which has helped to make our presentation more homogeneous.

Response to the Reviewer #2

Reviewer writes:

I have revised the new version of the manuscript. All my comments have been addressed and I therefore recommend publication in its current form.

Authors respond:

We thank the reviewer for the positive recommendation and for the constructive criticism in the previous rounds.

Response to the Reviewer #3

Editor writes:

Ref3 acknowledges that you correctly point out the analogy to localized Berry curvatures in crystalline systems but suggests to emphasize this from the very beginning and to consider recent works by M. Silveirinha on the issue of properly defining Chern numbers in such systems.

Authors respond:

We thank the reviewer for the positive opinion on our work. We are aware of the works by M. Silveirinha and we are happy to add several references to the introduction of our manuscript. While in our particular structure, because of the time-reversal symmetry the Chern number is zero, these references are still important to provide the global context. The phrase now reads:

Because of these two contributions, the photonic modes of a 2D continuous medium exhibit a non-zero Berry curvature [23,24], **with the possibility to define the associated topological invariants, such as the Chern numbers [25] and the Z2 invariant [26], and to observe the edge states and a non-zero angular momentum [27].**

The references 25-27 cite the key works of M. Silveirinha on the definition and the physical consequences of the topological invariants in continuous photonic systems.

Editor writes:

Ref3 also asks for a possible manifestation of OA induced topological textures like currents along domain walls.

Authors respond:

This is a very good question which has inspired us to make some quick calculations and to add the following text into the Discussion section of the manuscript:

With the Berry curvature localized in analogues of valleys which are well represented mathematically by tilted Dirac cones, it could be possible to create interface states of the Jackiw-Rebbi type [67] between regions of opposite topology defined by the sign of the emergent optical activity ζ , with a single chiral state for each "valley". These states exhibit valley-dependent group velocity, whose direction (in the reference frame of the tilted Dirac cone) can be inverted by changing the order of the topological materials, as in the quantum valley Hall effect [17,68]. This behavior, confirmed by our preliminary simulations, will be a subject of a separate future work.

Editor writes:

To enhance the readability ref3 suggests to discuss the difference between the TE-TM and the H-V modes and to discuss the physical origin of the finite beta.

Authors respond:

We have extended the discussion of the TE-TM and HV splittings in response to the comments of the Reviewer 1. Both contributions are discussed in detail in the paragraph containing Eq. (3). The added phrase reads:

The important difference between the TE-TM and the H-V splittings is that the orientation of TE-TM is defined by the in-plane wave vector (the orientation of the incidence plane with respect to the polarization-resolving detector), while the orientation of the H-V splitting is linked with the crystal axes and thus stays constant, because the sample is not rotated in our experiments.

In addition, we have added the following phrase, commenting the physical origin of the TE-TM splitting, described by the parameter beta:

The first contribution is the TE-TM splitting, ubiquitous in 2D photonic systems [24]. In general, the TE-TM splitting appears in any inhomogeneous system, in presence of any gradient allowing to define the transverse directions for the field [32]. In particular, in planar cavities it appears because of the polarization-dependent reflection coefficients [48].

We note that the details of the calculations of beta can be found in the two references [21,48], that we cite before Eq. (3). We use it as a fitting parameter, whose value is given at the end of the section Results.

We thank all reviewers and the Editor once again for their extremely constructive approach to the reviewing of our manuscript.

Sincerely yours,

The authors